# Defending against Adversarial Audio via Diffusion Model

**Shutong Wu**[1,2*]  **Jiongxiao Wang**[1]    **Wei Ping**[3]    **Weili Nie**[3]    **Chaowei Xiao**[1]
[1]Arizona State University    [2]Shanghai Jiao Tong University    [3]NVIDIA

## Abstract

Deep learning models have been widely used in commercial acoustic systems in recent years. However, adversarial audio examples can cause abnormal behaviors for those acoustic systems, while being hard for humans to perceive. Various methods, such as transformation-based defenses and adversarial training, have been proposed to protect acoustic systems from adversarial attacks, but they are less effective against adaptive attacks. Furthermore, directly applying the methods from the image domain can lead to suboptimal results because of the unique properties of audio data. In this paper, we propose an adversarial purification-based defense pipeline, AudioPure, for acoustic systems via off-the-shelf diffusion models. Taking advantage of the strong generation ability of diffusion models, AudioPure first adds a small amount of noise to the adversarial audio and then runs the reverse sampling step to purify the noisy audio and recover clean audio. AudioPure is a plug-and-play method that can be directly applied to any pretrained classifier without any fine-tuning or re-training. We conduct extensive experiments on speech command recognition task to evaluate the robustness of AudioPure. Our method is effective against diverse adversarial attacks (e.g. $\mathcal{L}_2$ or $\mathcal{L}_\infty$-norm). It outperforms the existing methods under both strong adaptive white-box and black-box attacks bounded by $\mathcal{L}_2$ or $\mathcal{L}_\infty$-norm (up to +20% in robust accuracy). Besides, we also evaluate the certified robustness for perturbations bounded by $\mathcal{L}_2$-norm via randomized smoothing. Our pipeline achieves a higher certified accuracy than baselines. Code is available at https://github.com/cychomatica/AudioPure.

## 1 Introduction

Deep neural networks (DNNs) have demonstrated great successes in different tasks in the audio domain, such as speech command recognition, keyword spotting, speaker identification, and automatic speech recognition. Acoustic systems built by DNNs (Amodei et al., 2016; Shen et al., 2019) are applied in safety-critical applications ranging from making phone calls to controlling household security systems. Although DNN-based models have exhibited significant performance improvement, extensive studies have shown that they are vulnerable to adversarial examples (Szegedy et al., 2014; Carlini & Wagner, 2018; Qin et al., 2019; Du et al., 2020; Abdullah et al., 2021; Chen et al., 2021a), where attackers add imperceptible and carefully crafted perturbations to the original audio to mislead the system with incorrect predictions. Thus, it becomes crucial to design robust DNN-based acoustic systems against adversarial examples.

To address it, existing works (e.g., Rajaratnam & Alshemali, 2018; Yang et al., 2019) have tried to leverage the temporal dependency property of audio to defend against adversarial examples. They apply the time-domain and frequency-domain transformations to the adversarial examples to improve the robustness. Although they can alleviate this problem to some extent, they are still vulnerable against strong adaptive attacks where the attacker obtains full knowledge of the whole acoustic system (Tramer et al., 2020). Another way to enhance the robustness against adversarial examples is adversarial training (Goodfellow et al., 2015; Madry et al., 2018) that adversarial perturbations have been added to the training stage. Although it has been acknowledged as the most effective defense, the training process will require expensive computational resources and the model is still

---

*work done during the internship at ASU.

vulnerable to other types of adversarial examples that are not similar to those used in the training process (Tramer & Boneh, 2019).

Adversarial purification (Yoon et al., 2021; Shi et al., 2021; Nie et al., 2022) is another family of defense methods that utilizes generative models to purify the adversarial perturbations of the input examples before they are fed into neural networks. The key of such methods is to design an effective generative model for purification. Recently, diffusion models have been shown to be the state-of-the-art models for images (Song & Ermon, 2019; Ho et al., 2020; Nichol & Dhariwal, 2021; Dhariwal & Nichol, 2021) and audio synthesis (Kong et al., 2021; Chen et al., 2021b). It motivates the community to use it for purification. In particular, in the image domain, DiffPure (Nie et al., 2022) applies diffusion models as purifiers and obtains good performance in terms of both clean and robust accuracy on various image classification tasks. Since such methods do not require training the model with pre-defined adversarial examples, they can generalize to diverse threats. Given the significant progress of diffusion models made in the image domain, it motivates us to ask: *is it possible to obtain similar success in the audio domain?*

Unlike the image domain, audio signals have some unique properties. There are different choices of audio representations, including raw waveforms and various types of time-frequency representations (e.g., Mel spectrogram, MFCC). When designing an acoustic system, some particular audio representations may be selected as the target features, and defenses that work well on some features may perform poorly on other features. In addition, one may think of treating the 2-D time-frequency representations (i.e., spectrogram) as images, where the frequency-axis is set as height and the time-axis is set as width, then directly apply the successful DiffPure (Nie et al., 2022) from the image domain for spectrogram. Despite the simplicity, there are two major issues: *i)* the acoustic system can take audio with variable time duration as the input, while the underlying diffusion model within DiffPure can only handle inputs with fixed width and height. *ii)* Even if we apply it in a fixed-length segment-wise manner for the time being, it still achieves the suboptimal results as we will demonstrate in this work. These unique issues pose a new challenge of designing and evaluating defense systems in the audio domain.

In this work, we aim to defend against diverse unseen adversarial examples without adversarial training. We propose a play-and-plug purification pipeline named AudioPure based on a pre-trained diffusion model by leveraging the unique properties of audio. In specific, our model consists of two main components: (1) a waveform-based diffusion model and (2) a classifier. It takes the audio waveform as input and leverages the diffusion model to purify the adversarial audio perturbations. Given an adversarial input formatted with waveform, AudioPure first adds a small amount of noise via the diffusion process to override the adversarial perturbations, and then uses the truncated reverse process to recover the clean sample. The recovered sample is fed into the classifier.

We conduct extensive experiments to evaluate the robustness of our method on the task of speech command recognition. We carefully design the adaptive attacks so that the attacker can accurately compute the full gradients to evaluate the effectiveness of our method. In addition, we also comprehensively evaluate the robustness of our method against different black-box attacks and the Expectation Over Transformation (EOT) attack. Our method shows a better performance under both white-box and black-box attacks against diverse adversarial examples. Moreover, we also evaluate the certified robustness of AudioPure via randomized smoothing, which offers a provable guarantee of model robustness against $\mathcal{L}_2$-based perturbation. We show that our method achieves better certified robustness than baselines. Specifically, our method obtains a significant improvement (up to +20% at most in robust accuracy) compared to adversarial training, and over 5% higher certified robust accuracy than baselines. To the best of our knowledge, we are the first to use diffusion models to enhance the security of acoustic systems and investigate how different working domains of defenses affect adversarial robustness.

## 2 RELATED WORK

**Adversarial attacks and defenses**. Szegedy et al. (2014) introduce adversarial examples, which look similar to normal examples but will fool the neural networks to give incorrect predictions. Usually, adversarial examples are constrained by $\mathcal{L}_p$ norm to ensure the imperceptibility. Recently, stronger attack methods are emerging (Madry et al., 2018; Carlini & Wagner, 2017; Andriushchenko et al., 2020; Croce & Hein, 2020; Xiao et al., 2018a;b; 2019; 2022b;a; Cao et al., 2019b;a; 2022a).

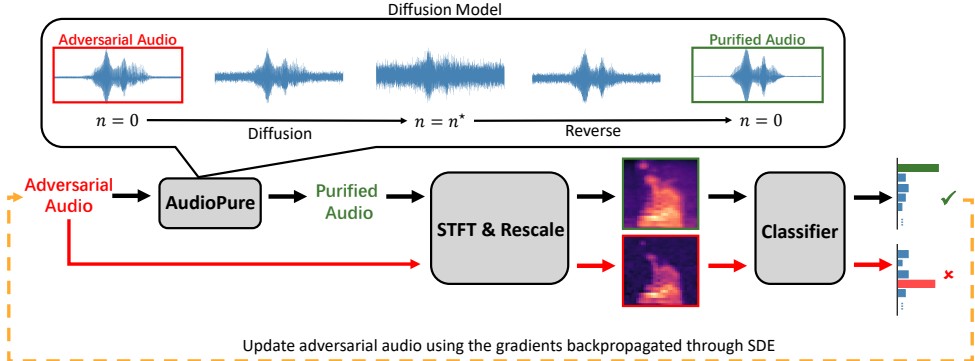

Figure 1: The architecture of the whole acoustic system protected by AudioPure (black line in the figure) and the adaptive attack (orange line in the figure). AudioPure first adds noise to the adversarial audio and then runs the reverse process to recover purified audio. Next, the purified audio is transformed into the spectrogram, and the spectrogram is fed into the classifier to get predictions. The attacker updates the adversarial audio based on the gradients backpropagated through SDE. Without AudioPure , the adversarial audio transfers to the spectrogram and feeds into the classifier directly.

In the audio domain, Carlini & Wagner (2018) introduce audio adversarial examples, and Qin et al. (2019) manage to make them more imperceptible. Black-box attacks (Du et al., 2020; Chen et al., 2021a) are also developed, aiming to mislead the end-to-end acoustic systems.

In order to protect neural networks from adversarial attacks, different defense methods are proposed. The most widely used one is adversarial training (Madry et al., 2018), which deliberately uses adversarial examples as the training data of neural networks. The main problems of adversarial training are the accuracy drop of benign examples and the expensive computational cost. Many improved versions of adversarial training aim to alleviate these problems (Wong et al., 2020; Shafahi et al., 2019; Zhang et al., 2019b;a; Sun et al., 2021; Cao et al., 2022b; Zhang et al., 2019c). Another line of work is adversarial purification (Yoon et al., 2021; Shi et al., 2021; Nie et al., 2022), which uses generative models to remove the adversarial perturbations before classification. Both of these two types of defenses are mainly developed for computer vision tasks and cannot be directly applied to the audio domain. In this paper, we explicitly design a defense pipeline according to the characteristics of audio data.

**Speech processing**. Many speech processing applications are vulnerable to adversarial attacks, including speech command recognition (Warden, 2018), keyword spotting (Chen et al., 2014; Li et al., 2019), speaker identification (Reynolds et al., 2000; Ravanelli & Bengio, 2018; Snyder et al., 2018), and speech recognition (Amodei et al., 2016; Shen et al., 2019; Ravanelli et al., 2019). In particular, speech command recognition is closely related to keyword spotting, and can be viewed as speech recognition with limited vocabulary. In this work, we choose speech command recognition as the testbed for the proposed AudioPure pipeline. The proposed pipeline is applicable for keyword spotting and speech recognition.

A speech command recognition system consists of a feature extractor and a classifier. The feature extractor processes the raw audio waveforms and outputs acoustic features, *e.g.* Mel spectrograms or Mel-frequency cepstral coefficients (MFCC). Then these features are fed into the classifier, and the classifier gives predictions. Given the 2-D spectrogram features, convolutional neural networks for images are readily applicable (Simonyan & Zisserman, 2015; He et al., 2016; Zagoruyko & Komodakis, 2016; Xie et al., 2017; Huang et al., 2017).

## 3 METHOD

### 3.1 BACKGROUND OF DIFFUSION MODELS

A diffusion model normally consists of a forward diffusion process and a reverse sampling process. The forward diffusion process gradually adds gaussian noise to the input data until the distribution

of the noisy data converges to a standard Gaussian distribution. The reverse sampling process takes the standard gaussian noise as input and gradually denoises the noisy data to recover clean data. At present, diffusion models can be divided into two different types: discrete-time diffusion models based on sequential sampling, such as SMLD Song & Ermon (2019), DDPM (Ho et al., 2020), and DDIM (Song et al., 2021a), and continuous-time diffusion models based on SDEs (Song et al., 2021c). Song et al. (2021c) also build the connection between these two types of diffusion models.

Denoising Diffusion Probabilistic Models (DDPM) (Ho et al., 2020) is one of the most widely used diffusion models. Many of the subsequently proposed diffusion models, including DiffWave for audio (Kong et al., 2021), are based on the DDPM formulation. In DDPM, both the diffusion and reverse processes are defined by Markov chains. For input data $\mathbf{x}_0 \in \mathbb{R}^d$, we denote $\mathbf{x}_0 \sim q(\mathbf{x}_0)$ as the original data distribution, and $\mathbf{x}_1, \ldots, \mathbf{x}_N$ are intermediate latent variables from the distributions $q(\mathbf{x}_1|\mathbf{x}_0), \ldots, q(\mathbf{x}_N|\mathbf{x}_{N-1})$, where $N$ is the total number of steps. Generally, with a pre-defined or learned variance schedule $\beta_1, \ldots, \beta_N$ (usually linearly increasing small constants), the forward transition probability $q(\mathbf{x}_n|\mathbf{x}_{n-1})$ can be formulated as:

$$q(\mathbf{x}_n|\mathbf{x}_{n-1}) = \mathcal{N}(\mathbf{x}_n; \sqrt{1-\beta_n}\mathbf{x}_{n-1}, \beta_n\mathbf{I}), \tag{1}$$

Based on the variance schedule $\{\beta_n\}$, a set of constants is defined as:

$$\alpha_n = 1 - \beta_n, \quad \bar{\alpha}_n = \prod_{n=1}^{N} \alpha_n, \quad \tilde{\beta}_n = \left\{ \begin{array}{ll} \frac{1-\bar{\alpha}_{n-1}}{1-\bar{\alpha}_n}\beta_n, & n > 1 \\ \beta_1, & n = 1 \end{array} \right., \tag{2}$$

and using the reparameterization trick, we have:

$$q(\mathbf{x}_n|\mathbf{x}_0) = \mathcal{N}(\mathbf{x}_n; \sqrt{\bar{\alpha}_n}\mathbf{x}_0, (1-\bar{\alpha}_n)\mathbf{I}) \tag{3}$$

When $n$ gradually gets larger to infinity, $q(\mathbf{x}_n|\mathbf{x}_0)$ will converge to a standard Gaussian distribution. Meanwhile, for the reverse process, we have:

$$\mathbf{x}_{n-1} \sim p_\theta(\mathbf{x}_{n-1}|\mathbf{x}_n) = \mathcal{N}(\mathbf{x}_{n-1}; \mu_\theta(\mathbf{x}_n, n), \sigma_\theta^2(\mathbf{x}_n, n)\mathbf{I}), \tag{4}$$

where the mean term $\mu_\theta(\mathbf{x}_n, n)$ and the variance term $\sigma_\theta^2(\mathbf{x}_n, n)$ is instantiated by parameter $\theta$. Ho et al. (2020); Kong et al. (2021) use a neural network $\epsilon_\theta$ to define $\mu_\theta$, and $\sigma_\theta$ is fixed to a constant:

$$\mu_\theta(\mathbf{x}_n, n) = \frac{1}{\sqrt{\alpha_n}}\left(\mathbf{x}_n - \frac{\beta_n}{\sqrt{1-\bar{\alpha}_n}}\epsilon_\theta(\mathbf{x}_n, n)\right), \quad \sigma_\theta(\mathbf{x}_n, n) = \sqrt{\tilde{\beta}_n}. \tag{5}$$

We denote $\mathbf{x}_n(\mathbf{x}_0, \epsilon) = \sqrt{\bar{\alpha}_n}\mathbf{x}_0 + \sqrt{(1-\bar{\alpha}_n)}\epsilon, \epsilon \sim \mathcal{N}(0, \mathbf{I})$, and the optimization objective is:

$$\theta^\star = \arg\max_\theta \sum_{n=1}^{N} \lambda_n \mathbb{E}_{\mathbf{x}(0)} \left\| \epsilon - \epsilon_\theta(\sqrt{\bar{\alpha}_n}\mathbf{x}_0 + \sqrt{(1-\bar{\alpha}_n)}\epsilon, n) \right\|_2^2 \tag{6}$$

where $\lambda_n$ is the weighting coefficient (Ho et al., 2020).

According to Song et al. (2021c), as $N \to \infty$, DDPM becomes VP-SDE, a continuous-time formulation of diffusion models. Particularly, the forward SDE is formulated as:

$$d\mathbf{x} = -\frac{1}{2}\beta(t)\mathbf{x}dt + \sqrt{\beta(t)}d\mathbf{w}. \tag{7}$$

where $t \in [0, 1]$, $dt$ is an infinitesimal positive time step, $\mathbf{w}$ is a standard Wiener process, $\beta(t)$ is the continuous-time noise schedule. Similarly, the reverse SDE can be defined as:

$$d\mathbf{x} = -\frac{1}{2}\beta(t)[\mathbf{x} + 2\nabla_\mathbf{x}\log p_t(\mathbf{x})]dt + \sqrt{\beta(t)}d\bar{\mathbf{w}}, \tag{8}$$

where $dt$ is an infinitesimal negative time step, and $\bar{\mathbf{w}}$ is a reverse-time standard Wiener process.

### 3.2 AUDIOPURE: A PLUG-AND-PLAY DEFENSE FOR ACOUSTIC SYSTEMS

To standardize the formulation of the defense, as suggested by Nie et al. (2022), we use the continuous-time formulation defined by Eq. 7 and Eq. 8. Note that since the existing pretrained DiffWave models (Kong et al., 2021) are based on DDPM, we will use their equivalent VP-SDE.

If we use the Euler-Maruyama method to solve the VP-SDE and the step size $\Delta t = \frac{1}{N}$, the sampling of the reverse-time SDE will be equivalent to the reverse sampling of DDPM (detailed proofs can be found in Song et al. (2021c)). Under this prerequisite, we have $t = \frac{n}{N}$ where $n \in \{1, \ldots, N\}$. We define $\beta(\frac{n}{N}) := \beta_n$, $\bar{\alpha}(\frac{n}{N}) := \bar{\alpha}_n$, $\tilde{\beta}(\frac{n}{N}) := \tilde{\beta}_n$, and $\mathbf{x}(\frac{n}{N}) := \mathbf{x}_n$. Given an adversarial example $x_{adv}$ as the input at $t = 0$, *i.e.* $\mathbf{x}_0 = \mathbf{x}_{adv}$, we first run the forward SDE from $t = 0$ to $t^\star = \frac{n^\star}{N}$ by solving Eq. 7 (it is equivalent to running $n^*$ DPPM steps), which yields:

$$\mathbf{x}(t^\star) = \sqrt{\bar{\alpha}(t^\star)}\mathbf{x}_{adv} + \sqrt{1 - \bar{\alpha}(t^\star)}\mathbf{z}, \qquad \mathbf{z} \sim \mathcal{N}(0, \mathbf{I}), \tag{9}$$

Next, we run the truncated reverse SDE from $t = t^\star$ to $t = 0$ by solving Eq. 8. Similar to Nie et al. (2022), we define an SDE solver **sdeint** that uses the Euler-Maruyama method, and sequentially takes in six inputs: initial value, drift coefficient, diffusion coefficient, Wiener process, initial time, and end time. The reverse output $\hat{\mathbf{x}}(0)$ at $t = 0$ can be formulated as:

$$\hat{\mathbf{x}}(0) = \mathbf{sdeint}(\mathbf{x}(t^\star), f_{rev}, g_{rev}, \bar{\mathbf{w}}, t^\star, 0). \tag{10}$$

where the drift and diffusion coefficients are:

$$f_{rev}(\mathbf{x}, t) := -\frac{1}{2}\beta(t)[\mathbf{x} + 2\mathbf{s}_\theta(\mathbf{x}, t)], \qquad g_{rev}(t) := \sqrt{\tilde{\beta}(t)}. \tag{11}$$

Note that we use a diffusion coefficient different from Nie et al. (2022) for the purpose of cleaner output (see the detailed explanation in Section 3.3). Next, we use the discrete-time noise estimator $\epsilon_\theta(\mathbf{x}_n, n)$ to compute the continuous-time score estimator $s_\theta(\mathbf{x}, t)$. By defining $\tilde{\epsilon}_\theta(\mathbf{x}(t), t) := \epsilon_\theta(\mathbf{x}(\frac{n}{N}), n) = \epsilon_\theta(\mathbf{x}_n, n)$ with $t := \frac{n}{N}$, the score function in the reverse VP-SDE can be estimated as:

$$\mathbf{s}_\theta(\mathbf{x}, t) = -\frac{\tilde{\epsilon}_\theta(\mathbf{x}, t)}{\sqrt{1 - \bar{\alpha}(t)}} \approx \nabla_\mathbf{x} \log p_t(\mathbf{x}). \tag{12}$$

Accordingly, $\hat{\mathbf{x}}(0)$, the purified output of the adversarial input $\mathbf{x}(0) = \mathbf{x}_{adv}$, is fed into the later stages of the acoustic system to make predictions. The whole purification operation can be denoted as a function $\mathbf{Purifier} : \mathbb{R}^d \times \mathbb{R} \to \mathbb{R}^d$:

$$\mathbf{Purifier}(\mathbf{x}_{adv}, n^\star) = \mathbf{sdeint}\left(\sqrt{\bar{\alpha}(\frac{n^\star}{N})}\mathbf{x}_{adv} + \sqrt{1 - \bar{\alpha}(\frac{n^\star}{N})}\mathbf{z}, f_{rev}, g_{rev}, \bar{\mathbf{w}}, \frac{n^\star}{N}, 0\right) \tag{13}$$

The acoustic systems are usually built on the features extracted from the raw audio. For example, the system can extract Mel spectrogram as the features: 1) it first applies short-time Fourier transformation (STFT) on the time-domain waveform to get linear-scale spectrogram, and 2) it then rescales the frequency band to the Mel-scale. We denote this process as Wave2Mel : $\mathbb{R}^d \to \mathbb{R}^m \times \mathbb{R}^n$, which is a differentiable function. Then the classifier $F : \mathbb{R}^m \times \mathbb{R}^n \to \mathbb{R}^c$ (usually a convolutional network) takes the Mel spectrogram as the input and gives predictions.

Since both the time domain waveform and time-frequency domain spectrogram go through the pipeline, the purifier can be applied in either the time domain or time-frequency domain. If the purifier is applied in the time domain, the whole defended acoustic system $\mathbf{AS} : \mathbb{R}^d \times \mathbb{R} \to \mathbb{R}^c$ can be formulated as:

$$\mathbf{AS}(\mathbf{x}_{adv}, n^\star) = F(\text{Wave2Mel}(\mathbf{Purifier}(\mathbf{x}_{adv}, n^\star))) \tag{14}$$

where the waveform $\mathbf{Purifier}$ is based on DiffWave.

Meanwhile, if we want to purify the input adversarial examples in the time-frequency domain, we can choose a diffusion model used for image synthesis, and apply it to the output spectrogram of Wave2Mel. We denote this purifier as $\mathbf{Purifier}_{spec} : \mathbb{R}^m \times \mathbb{R}^n \times \mathbb{R} \to \mathbb{R}^m \times \mathbb{R}^n$. In this scenario, the whole defended acoustic system will be:

$$\mathbf{AS}(\mathbf{x}_{adv}, n^\star) = F(\mathbf{Purifier}_{spec}(\text{Wave2Mel}(\mathbf{x}_{adv})), n^\star) \tag{15}$$

The architecture of the whole pipeline is illustrated in Figure. 1. For the purification in the time-frequency domain spectrogram, we use an Improved DDPM (Nichol & Dhariwal, 2021) trained on the Mel spectrograms of audio data and denote it as *DiffSpec*. We compare these two purifiers and discover that the purification in the time domain waveform is more effective to defend against adversarial audio. Detailed experimental results can be found in Sec. 4.2.

### 3.3 Towards evaluating AudioPure

**Adaptive attack** For the forward diffusion process formulated as Eq. 9, the gradients of the output $\mathbf{x}(t^\star)$ *w.r.t.* the input $\mathbf{x}(0)$ is a constant. For the reverse process formulated as Eq. 10, the adjoint method (Li et al., 2020) is applied to compute the full gradients of the objective function $\mathcal{L}$ *w.r.t.* $x(t^\star)$ without any out-of-memory issues, by solving another augmented SDE:

$$\begin{pmatrix} \mathbf{x}(t^\star) \\ \frac{\partial \mathcal{L}}{\partial \mathbf{x}(t^\star)} \end{pmatrix} = \mathbf{sdeint} \left( \begin{pmatrix} \mathbf{x}(0) \\ \frac{\partial \mathcal{L}}{\partial \mathbf{x}(0)} \end{pmatrix}, \begin{pmatrix} f_{rev} \\ \frac{\partial f_{rev}}{\partial \mathbf{x}} \mathbf{z} \end{pmatrix}, \begin{pmatrix} g_{rev}\mathbf{1} \\ \mathbf{0} \end{pmatrix}, \begin{pmatrix} -\mathbf{w}(1-t) \\ -\mathbf{w}(1-t) \end{pmatrix}, 0, t^\star \right) \quad (16)$$

where $\mathbf{1}$ and $\mathbf{0}$ represent the vectors of all ones and all zeros, respectively.

**SDE modifications for clean output** We observe that directly applying the framework of Nie et al. (2022) to the audio domain will cause the performance degradation. That is, when converting the discrete-time reverse process of DiffWave (Kong et al., 2021) to its corresponding reverse VP-SDE in Eq. 8, the output audio still contains much noise, resulting in lower classification accuracy. We identify two influencing factors and solve this problem by modifying the SDE formulation.

The first factor is the diffusion error due to the mismatch of the reverse variance between the discrete and continuous cases. Ho et al. (2020) observed that both $\sigma_\theta^2 = \tilde{\beta}_t$ and $\sigma_\theta^2 = \beta_t$ get similar results experimentally in the image domain. However, we find that it is not the case in the audio modeling with diffusion models. For audio synthesis using DiffWave trained with $\sigma_\theta^2 = \tilde{\beta}_t$, if we switch the reverse variance schedule to $\sigma_\theta^2 = \beta_t$, the output audio becomes noisy. Thus, in Sec. 3.2 we define $\tilde{\beta}(\frac{n}{N}) = \tilde{\beta}_n$ and use the diffusion coefficient $g_{rev} = \sqrt{\tilde{\beta}(t)}$ in Eq. 11 instead of $g_{rev} = \sqrt{\beta(t)}$ to match the variance $\tilde{\beta}_t$ in DiffWave.

The second factor is the inaccuracy from the continuous-time noise schedule $\beta(t) = \beta_0 + (\beta_N - \beta_0)t$ and $\tilde{\alpha}(t) = e^{-\int_0^t \beta(s)\mathrm{d}s}$ used by Nie et al. (2022). The impact of the difference between $\beta(t) = \beta_0 + (\beta_N - \beta_0)t$ and $\beta_N t$ cannot be negligible, especially when $N$ is not large enough (*e.g.* $N = 200$ for the pretrained DiffWave model we use). Besides, when $t$ is close to 0, $\tilde{\alpha}(t) = e^{-\int_0^t \beta(s)\mathrm{d}s}$ is not a good approximation of $\bar{\alpha}_{Nt}$ any more. Thus, we define the continuous-time noise schedule directly based on the discrete schedule, namely, $\beta(\frac{n}{N}) := \beta_n$ and $\bar{\alpha}(\frac{n}{N}) := \bar{\alpha}_n$, for the purpose of better denoised output and more accurate gradient computation.

## 4 Experiments

In this section, we first introduce the detailed experimental settings. Then we compare the performance of our method and other defenses under strong white-box adaptive attack where the attacker has full knowledge about the defense and black-box attacks. To further show the robustness of our method, we also evaluate the certified accuracy via randomize smoothing Cohen et al. (2019), which provides a provable guarantee of model robustness against $\mathcal{L}_2$ norm bounded adversarial perturbations.

### 4.1 Experimental settings

**Dataset**. Our method is evaluated on the task of speech command recognition. We use the Speech Commands dataset (Warden, 2018), which consists of 85,511 training utterances, 10,102 validation utterances, and 4,890 tests utterances. Following the setting of Kong et al. (2021), we choose the utterances which stand for digits $0 \sim 9$ and denote this subset as SC09.

**Models**. We use DiffWave (Kong et al., 2021) and DiffSpec (based on Improved DDPM (Nichol & Dhariwal, 2021)) as our defensive purifiers, which are representative diffusion models on the waveform domain and the spectral domain respectively. We use the unconditional version of Diffwave with the officially provided pretrained checkpoints. Since the Improved DDPM model does not provide the pretrained checkpoint for audio, we train it from scratch on the Mel spectrograms of audio from SC09. The training details and hyperparameters are in the appendix A. For the classifier, we use ResNeXt-29-8-64(Xie et al., 2017) for spectrogram representation and M5 Net(Dai et al., 2017) for waveform except the experiments for ablation studies.

Table 1: Performance against adaptive attacks among different methods.

| Defense | Clean | $\mathcal{L}_\infty$ white-box | | | | | | $\mathcal{L}_2$ white-box | | | | | $\mathcal{L}_\infty$ black-box |
|---|---|---|---|---|---|---|---|---|---|---|---|---|---|
| | | PGD$_{10}$ | PGD$_{20}$ | PGD$_{30}$ | PGD$_{50}$ | PGD$_{70}$ | PGD$_{100}$ | PGD$_{10}$ | PGD$_{20}$ | PGD$_{30}$ | PGD$_{50}$ | PGD$_{100}$ | FAKEBOB |
| None | **100** | 3 | 1 | 1 | 1 | 1 | 1 | 2 | 0 | 0 | 0 | 0 | 21 |
| AS (Yang et al., 2019) | **100** | 4 | 2 | 1 | 1 | 1 | 1 | 1 | 0 | 0 | 0 | 0 | 24 |
| MS (Yang et al., 2019) | **100** | 6 | 3 | 2 | 1 | 1 | 1 | 4 | 1 | 0 | 0 | 0 | 21 |
| DS (Yang et al., 2019) | 99 | 2 | 1 | 1 | 1 | 1 | 1 | 0 | 0 | 0 | 0 | 0 | 16 |
| LPF (Rajaratnam & Alshemali, 2018) | **100** | 5 | 2 | 1 | 1 | 1 | 1 | 2 | 0 | 0 | 0 | 0 | 20 |
| BPF (Rajaratnam & Alshemali, 2018) | 99 | 5 | 1 | 1 | 1 | 1 | 1 | 1 | 1 | 0 | 0 | 0 | 18 |
| AdvTr (Madry et al., 2018) | **100** | 86 | 79 | 78 | 74 | 72 | 71 | 73 | 70 | 68 | 65 | 65 | **92** |
| AudioPure | 97 | **89** | **89** | **89** | **85** | **84** | **84** | **89** | **86** | **83** | **85** | **84** | 86 |

**Attacks**. For white-box attacks, we use PGD (Madry et al., 2018) with different iteration steps from 10 to 100 among $L_\infty$ and $L_2$ norms. The attack budget is set to $\epsilon = 0.002$ for $\mathcal{L}_\infty$-norm constraint except the ablation study and $\epsilon = 0.253$ for $L_2$ norm constraint. For black-box attacks, we apply a query-based attack, FAKEBOB (Chen et al., 2021a), and set the iteration steps to 200, NES samples to 200, and the confidence coefficient $\kappa = 0.5$.

**Baselines**. We compare our method with two types of baselines including: (1) transformation-based defense (Yang et al., 2019; Rajaratnam & Alshemali, 2018) including average smoothing (AS), median smoothing (MS), downsampling (DS), low-pass filter (LPF), and band-pass filter (BPF), and (2) adversarial training based defense (AdvTr) (Madry et al., 2018). For adversarial training, we follow the setting of Chen et al. (2022), using $\mathcal{L}_\infty$ PGD$_{10}$ with $\epsilon = 0.002$ and $ratio = 0.5$.

## 4.2 MAIN RESULTS

We evaluate AudioPure ($n^\star$=3 by default) under adaptive attacks, assuming the attacker obtains full knowledge of our defense. We use the adaptive attack algorithm described in the previous section so that the attacker is able to accurately compute the full gradients for attacking. The results are shown in Table 1. We find that the baseline transformation-based defenses (Yang et al., 2019; Rajaratnam & Alshemali, 2018), including average smoothing (AS), median smoothing (MS), downsampling (DS), low-pass filter (LPF), and band-pass filter (BPF), are virtually broken through by white-box attacks with up to 4% robust accuracy. For the adversarial training-based method (AdvTr) trained on $\mathcal{L}_\infty$-norm adversarial examples, although it achieves 71% robust accuracy against $\mathcal{L}_\infty$-based adversarial examples, such the method does not work so well on other types of adversarial examples (i.e., $\mathcal{L}_2$-based method), achieving 65% robust accuracy under $\mathcal{L}_2$-based PDG$_{100}$ attack. Compared with all baselines, AudioPure can obtain much higher robust accuracy, about 10% improvements on average, on $\mathcal{L}_\infty$-based adversarial examples, and is equally effective against $\mathcal{L}_2$-based white-box attacks, achieving 84% robust accuracy.

We also evaluate AudioPure on black-box attacks including: FAKEBOB (Chen et al., 2021a) and transferability-based attacks. The results of FAKEBOB are shown in Table 1, indicating that our method can keep effective under the query-based black-box attack. The results of the transferability-based attack are in the appendix B. They draw the same conclusion. These results further verify the effectiveness of our method. All results indicate that AudioPure can work under diverse attacks with different types of constraints, while adversarial training has to apply different training strategies and re-train the model, making it less effective among unseen attacks than our method. We report the actual inference time in Appendix J and compare out method with more existing methods in Appendix F, G and H. Additionally, we conduct experiments on the Qualcomm Keyword Speech Dataset (Kim et al., 2019), and the results and details are in Appendix E. In this dataset, our method is still effective against adversarial examples.

## 4.3 ABLATION STUDY

**PGD steps** To ensure the effectiveness of PGD attacks, we test different iteration steps from 10 to 150. As Figure. 2a illustrates, the robust accuracy converges after iteration steps $n \geq 70$.

**Effectiveness against Expectation over transformation (EOT) attack.** Besides, since the diffusion and reverse process of AudioPure consist of many randomized sampling operations, we apply the expectation over transformation (EOT) attack to evaluate the effectiveness of AudioPure with

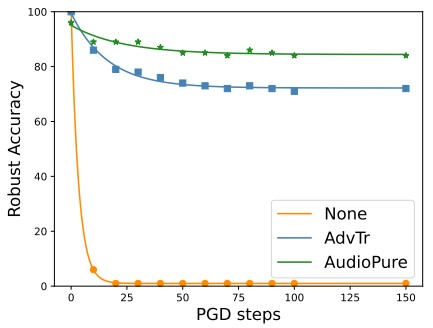
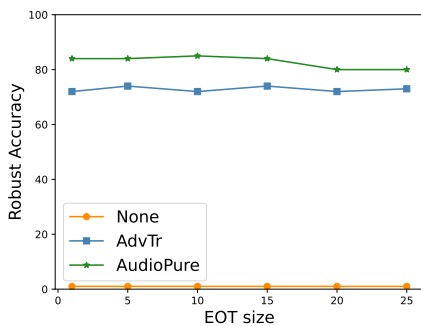

(a) robust accuracy with different PGD steps  (b) robust accuracy with different EOT size

Figure 2: The performance of baseline (no defense, denoted as None), adversarial training (denoted as AdvTr), and AudioPure under attacks with different iteration steps and EOT size. (a) indicates the step of convergence, and the attack is almost optimal when iterating over 70 steps. (b) shows that increasing EOT size can barely affect the robustness of our method.

Table 2: The robust accuracy under $PGD_{10}$ with different attack budget $\epsilon$ when using different reverse steps $n^{\star}$. Larger $\epsilon$ requires larger $n^{\star}$ to ensure better robustness.

| Attack Budget | Diffusion Steps | | | | | | |
|---|---|---|---|---|---|---|---|
| | $n^{\star} = 0$ | $n^{\star} = 1$ | $n^{\star} = 2$ | $n^{\star} = 3$ | $n^{\star} = 5$ | $n^{\star} = 7$ | $n^{\star} = 10$ |
| $\epsilon = 0.002$ | 3 | **94** | 90 | 89 | 84 | 77 | 67 |
| $\epsilon = 0.004$ | 0 | 76 | **89** | 86 | 83 | 74 | 66 |
| $\epsilon = 0.008$ | 0 | 27 | 70 | **85** | 84 | 74 | 68 |
| $\epsilon = 0.016$ | 0 | 0 | 21 | 53 | **69** | 57 | 63 |

Table 3: Ablation studies among different model architectures. The robust accuracy is evaluated under $\mathcal{L}_{\infty}$-$PGD_{70}$. Our method is effective on various models with different architectures.

| Defense | ResNeXt-29-8-64 | | VGG-19-BN | | WideResNet-28-10 | | DenseNet-BC-100-12 | | M5 | |
|---|---|---|---|---|---|---|---|---|---|---|
| | Clean | Robust | Clean | Robust | Clean | Robust | Clean | Robust | Clean | Robust |
| None | 100 | 1 | 100 | 2 | 100 | 1 | 100 | 5 | 94 | 12 |
| AudioPure | 97 | 84 | 99 | 81 | 99 | 85 | 96 | 79 | 94 | 70 |

different EOT sample sizes. Figure 2b demonstrates the result. We find that AudioPure is effective among different EOT sizes.

**Attack budget $\epsilon$.** We evaluate the effectiveness of our method among different $\epsilon$ including $\epsilon = \{0.002, 0.004, 0.008, 0.016\}$. Since the diffusion steps $n^{\star}$ are the hyperparameters for AudioPure, we conduct experiments among different $n^{\star}$. As shown in Table 2, if $n^{\star}$ is larger than 2, AudioPure will show strong effectiveness among different $\epsilon$. When $\epsilon$ increases, it requires a larger $n^{\star}$ to achieve the optimal robustness since a larger adversarial perturbation requires a large noise from the forward process of the diffusion model to override the adversarial perturbations and the corresponding larger step to recover purified audio. However, if the $n^{\star}$ is too large, it will override the original audio information as well so that the recovered audio from the diffusion model will lose the original audio information, contributing to the performance drop. Furthermore, we explore the extent of the diffusion model for purification in Appendix I.

**Architectures.** Moreover, we apply AudioPure to different classifiers, including spectrogram-based classifier: VGG-19-BN(Simonyan & Zisserman, 2015), ResNeXt-29-8-64(Xie et al., 2017), WideResNet-28-10(Zagoruyko & Komodakis, 2016) DenseNet-BC-100-12(Huang et al., 2017) and wave-form based classifier: M5 (Dai et al., 2017). Table 3 shows that our method is effective for various neural network classifiers.

**Audio representations** Audio has different types of representations including raw waveforms or time-frequency representations (e.g., Mel spectrogram). We conduct an ablation study to show the effectiveness of diffusion models by using different representations, including *DiffWave*, a diffusion model for waveforms (Kong et al., 2021) and *DiffSpec*, a diffusion model for spectrogram based on the original image model (Nichol & Dhariwal, 2021). The results are shown in Table 4. We find

Table 4: Ablation studies among different audio representations. We implement AudioPure using two different diffusion models as purifiers, DiffWave and DiffSpec, that respectively process the representations in the time domain and time-frequency domain.

| Defense | Clean | $\mathcal{L}_\infty$ white-box | | | | | | $\mathcal{L}_2$ white-box | | | | |
|---------|-------|-------------|-------------|-------------|-------------|-------------|--------------|-------------|-------------|-------------|-------------|--------------|
| | | $PGD_{10}$ | $PGD_{20}$ | $PGD_{30}$ | $PGD_{50}$ | $PGD_{70}$ | $PGD_{100}$ | $PGD_{10}$ | $PGD_{20}$ | $PGD_{30}$ | $PGD_{50}$ | $PGD_{100}$ |
| DiffWave | 97 | 89 | **89** | **89** | **85** | **84** | **84** | **89** | **86** | **83** | **85** | **84** |
| DiffSpec | **99** | **92** | 84 | 78 | 75 | 72 | 71 | 74 | 62 | 58 | 54 | 49 |

Table 5: Certified accuracy for different methods. For each noise level $\sigma$, we add the same level of noise to train the classifier and apply it to RS-Gaussian.

| Method | Noise level | Certified radius ($\mathcal{L}_2$) | | | | | | |
|--------|-------------|-----|------|------|------|-----|------|------|
| | | 0 | 0.25 | 0.50 | 0.75 | 1.0 | 1.25 | 1.50 |
| RS-Vanilla | $\sigma = 0.5$ | 30 | 21 | 12 | 6 | 4 | 3 | 3 |
| | $\sigma = 1.0$ | 8 | 8 | 8 | 7 | 4 | 3 | 3 |
| RS-Gaussian | $\sigma = 0.5$ | **49** | 39 | 33 | 23 | 14 | 6 | 3 |
| | $\sigma = 1.0$ | 18 | 15 | 11 | 10 | 5 | 5 | 4 |
| AudioPure | $\sigma = 0.5$ | 45 | **40** | **35** | **27** | **21** | **17** | **13** |
| | $\sigma = 1.0$ | **27** | **22** | **16** | **15** | **12** | **11** | **8** |

that the *DiffWave* consistently outperforms *DiffSpec* against $\mathcal{L}_2$ and $\mathcal{L}_\infty$-based adversarial examples. Moreover, compared with *DiffWave*, despite *DiffSpec* achieve higher clean accuracy, it only achieves 49% robust accuracy, a significant 35% performance drop against $\mathcal{L}_2$-based adversarial examples. We think the potential reason is that the short-time Fourier transform (STFT) is an operation of information compression. The spectrogram contains much less information than the raw audio waveform. This experiment shows that the domain difference contributes to significantly different results, and directly applying the method from the image domain can lead to suboptimal performance for audio. It also verifies the crucial design of AudioPure for adversarial robustness.

## 4.4 CERTIFIED ROBUSTNESS

In this section, we evaluate the certified robustness of AudioPure via randomized smoothing(Cohen et al., 2019). Here we draw $N = 100,000$ noise samples and select noise levels $\sigma \in \{0.5, 1.0\}$ for certification. Note that we follow the same setting from Carlini et al. (2022) and choose to use the one-shot denoising method. The detailed implementation of our method could be found in Appendix C. We compare our results with randomized smoothing using the vanilla classifier and Gaussian augmented classifier, denoted RS-Vanilla and RS-Gaussian respectively. The results are shown in Table 5. We also provide the certified robustness under different $\mathcal{L}_2$ perturbation budget with different Gaussian noise $\sigma = \{0.5, 1.0\}$ in Figure A of Appendix C. By observing our results, we find that our method outperforms baselines for a better certified accuracy except $\sigma = 0.5$ at 0 radius. We also notice that the performance of our method will be even better when the input noise gets larger. This may be due to AudioPure can still recover the clean audio with a large $\mathcal{L}_2$-based perturbation while Gaussian augmented model could even not be converged when training with such large noise.

## 5 CONCLUSION

In this paper, we propose an adversarial purification-based defense pipeline for acoustic systems. To evaluate the effectiveness of AudioPure , we design the adaptive attack method and evaluate our method among adaptive attacks, EOT attacks, and black-box attacks. Comprehensive experiments indicate that our defense is more effective than existing methods (including adversarial training) among the diverse type of adversarial examples. We show AudioPure achieves better certifiable robustness via Randomized Smoothing than other baselines. Moreover, our defense can be a universal plug-and-play method for classifiers with different architectures.

**Limitations.** AudioPure introduces the diffusion model, which increases the time and computational cost. Thus, how to improve time and computational efficiency is an important future work. For example, it is interesting to investigate the distillation technique (Salimans & Ho, 2022) and fast sampling method (Kong & Ping, 2021) to reduce the computation complexity introduced by diffusion models.

ACKNOWLEDGMENT

We thank Prof. Xiaolin Huang from Shanghai Jiao Tong University for the valuable discussions. Shutong Wu is partially supported by the National Natural Science Foundation of China (61977046), Shanghai Science and Technology Program (22511105600), and Shanghai Municipal Science and Technology Major Project (2021SHZDZX0102).

ETHICS STATEMENT

Our work proposes a defense pipeline for protecting acoustic systems from adversarial audio examples. In particular, our study focuses on speech command recognition, which is closely related to keyword spotting systems. Such systems are well known to be vulnerable to adversarial attacks. Our pipeline will enhance the security aspect of such real-world acoustic systems and benefit the social beings. The Speech Commands dataset used in our study are released by others and has been publicly available for years. The dataset contains various voices from anonymous speakers. To the best of our knowledge, it does not contain any privacy-related information for these speakers.

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

# APPENDIX

## A DETAILS ON TRAINING THE IMPROVE DDPM.

We train an Improved DDPM using the official repository (https://github.com/openai/improved-diffusion). For the UNet model, we set $image\_size = 32$, $num\_channels = 3$, and $num\_res\_blocks = 128$. For diffusion flags, we set $N = 200$, $\beta_1 = 0.0001$, $\beta_N = 0.02$ and use the linear variance schedule. For the model training, we set the learning rate to $1e - 4$ and the batch size to 230. The training loss has converged after 80,000 training steps, and we use this checkpoint to build our purifier.

## B ADDITIONAL EXPERIMENTS OF TRANSFER-BASED ATTACK

We additionally evaluate our method under transfer-based attack, where we assume the attacker can only get the output logits of the acoustic system but have no knowledge about the used defense.

We use model functional stealing to train a surrogate model. Specifically, we first feed input examples into the acoustic system consisting of DiffWave and a ResNeXt classifier and get the output logits. Then we use these output logits of the acoustic system as labels and train a new surrogate ResNeXt model, which has the same architecture as the classifier in the acoustic system. The results are shown in Table A. The *Stealing Acc.* denotes the accuracy of the surrogate classifier using the predictions of the defended acoustic system as ground truth. The *Transfer to Vanilla* and *Transfer to Defended* represent the undefended vanilla classifier and the defended acoustic system. The surrogate classifier is attacked to generate adversarial examples, and these adversarial examples are transferred to evaluate the robustness of the undefended vanilla classifier and the defended acoustic system.

Table A: Transfer-based attack via model functional stealing. We train a surrogate model, using the outputs of the defended acoustic system as labels. Then adversarial examples are generated by attacking the surrogate model and transferred to the undefended vanilla classifier and the defended acoustic system.

| Stealing Target | Stealing Acc. | Transfer to Vanilla | | Transfer to Defended | |
| --- | --- | --- | --- | --- | --- |
| | | Clean | Robust | Clean | Robust |
| AudioPure ($n^\star = 1$) | 100 | 100 | 22 | 100 | 99 |
| AudioPure ($n^\star = 5$) | 98 | 100 | 58 | 96 | 94 |

## C DETAILS ABOUT CERTIFIED ROBUSTNESS

Randomized smoothing (Cohen et al., 2019) provides a provable robustness guarantee in $\mathcal{L}_2$-norm by evaluating models under noise. Usually, the performance of the vanilla classifier will degrade when feeding the Gaussian perturbed inputs. To alleviate this problem, we can re-train a new network or fine-tune a pretrained network on Gaussian augmented data. However, both of them could take a lot of time on training. Another way is to apply a denoiser before the vanilla classifier, named denoised smoothing (Salman et al., 2020). Since the reverse process of the diffusion model can be seen as a good denoiser, we can use a pretrained diffusion model as a plug-and-play method to make any model certifiably robust. For a given noise level $\sigma$, we can compute the corresponding diffusion step $t^\star$ which adds the same level of noise to the input examples. The diffusion process can be reformulated as:

$$\mathbf{x}_n = \sqrt{\bar{\alpha}_n}\mathbf{x}_0 + \sqrt{1 - \bar{\alpha}_t}\mathbf{z} = \sqrt{\bar{\alpha}_n}(\mathbf{x}_0 + \sqrt{\frac{1 - \bar{\alpha}_n}{\bar{\alpha}_n}}\mathbf{z}), \qquad \mathbf{z} \sim \mathcal{N}(0, \mathbf{I}), \qquad \text{(S17)}$$

while the noisy input $\hat{\mathbf{x}}$ of randomized smoothing is

$$\hat{\mathbf{x}} = \mathbf{x}_0 + \sqrt{\sigma}\mathbf{z}, \qquad \mathbf{z} \sim \mathcal{N}(0, \mathbf{I}). \qquad \text{(S18)}$$

So we can obtain $n^\star$ *s.t.* $\frac{1-\bar{\alpha}_n}{\bar{\alpha}_n} = \sigma$ after multiplying a rescale coefficient $\sqrt{\bar{\alpha}_n}$ on the input $\hat{\mathbf{x}}$.

According to Carlini et al. (2022), a single reverse step is able to recover an image with a high accuracy for the classifier and can largely save computational time by directly recovering the data through $\mathbf{x}_0 = \frac{1}{\sqrt{\bar{\alpha}_n}}(\mathbf{x}_n - \sqrt{1-\bar{\alpha}_n}\epsilon_\theta(\sqrt{\bar{\alpha}_n}\hat{\mathbf{x}}, n))$. So we can just apply one-shot denoising instead of running full steps in our reverse process.

Figure A shows the certified accuracy of AudioPure compared with RS-Gaussian and RS-Vanilla. The results show that the certified robustness of our method is consistently better than baselines except at small certified radii when $\sigma = 0.50$.

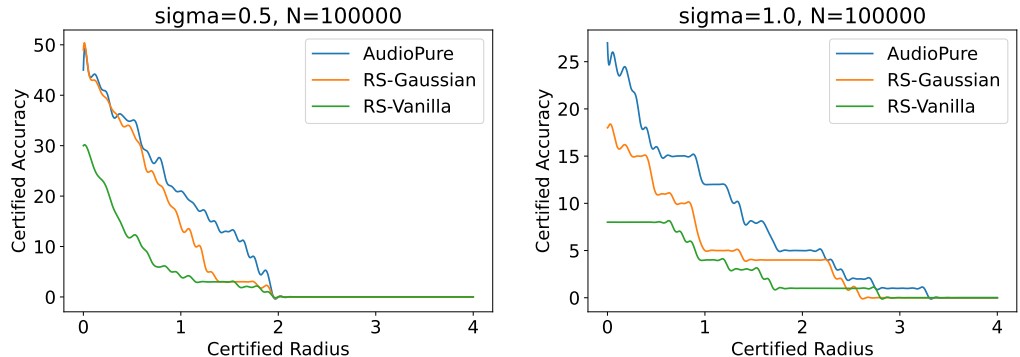

Figure A: Certified robustness ($\mathcal{L}_2$) with different input noise level $\sigma$. Larger $\sigma$ ensures better robustness under larger perturbations, but the performance for benign inputs will be degraded.

## D  THEORETICAL ANALYSIS ON THE PURIFICATION ABILITY

**Theorem D. 1**. Assume that $p(x)$ and $q(x)$ are respectively the data distribution of clean examples and the data distribution of adversarial examples. We use $p_t$ and $q_t$ to represent the respective distribution of $x(t)$ when $x(t) \sim p(x)$ and $x(t)$ when $x(t) \sim q(x)$. Then we have

$$\frac{\partial D_{KL}(p_t||q_t)}{\partial t} \leq 0 \tag{S19}$$

where the equality is established only if $p_t = q_t$. This inequality indicates that as $t$ increases from 0 to 1, the KL divergence of $p_t$ and $q_t$ monotonically decreases. In other words, when the diffusion steps $n^\star$ increases, more of the adversarial perturbations will be removed. Considering that the original semantic information will also be removed if $n^\star$ is too large, which affects the clean accuracy, there should be a trade-off when we set $n^\star$ for the diffusion model purifier.

**Proof**: Following Nie et al. (2022); Song et al. (2021b), we firstly formulate the Fokker-Planck equation (Särkkä & Solin, 2019) of the forward SDE in Eq. 7 (where we define $f(x,t) := -\frac{1}{2}\beta(t)$ and $g(t) := \sqrt{\beta(t)}$) as:

$$
\begin{aligned}
\frac{\partial p_t(x)}{\partial t} &= -\nabla_x\left(f(x,t)p_t(x) - \frac{1}{2}g^2(t)\nabla_x p_t(x)\right) \\
&= -\nabla_x\left(f(x,t)p_t(x) - \frac{1}{2}g^2(t)p_t(x)\nabla_x \log p_t(x)\right) \\
&= \nabla_x \cdot (h_p(x,t)p_t(x))
\end{aligned}
\tag{S20}
$$

where $h_p(x,t) := \frac{1}{2}g^2(t)\nabla_x \log p_t(x) - f(x,t)$. Assuming $p_t$ and $q_t$ are smooth and fast decaying, *i.e.* for any $i = 1, \ldots, d$, we have

$$\lim_{x_i\to\infty} p_t(x)\frac{\partial}{\partial x_i}\log p_t(x) = 0, \qquad \lim_{x_i\to\infty} q_t(x)\frac{\partial}{\partial x_i}\log q_t(x) = 0 \tag{S21}$$

for $x_i$, the $i$-th dimension of $x \in \mathbb{R}^d$. Then we reformulate the KL divergence as

$$
\begin{aligned}
\frac{\partial D_{KL}(p_t||q_t)}{\partial t} &= -\frac{\partial}{\partial t} \int p_t(x) \log \frac{p_t(x)}{q_t(x)} \mathrm{d}x \\
&= -\nabla_x \left( f(x,t)p_t(x) - \frac{1}{2}g^2(t)p_t(x)\nabla_x \log p_t(x) \right) \\
&= \int \nabla_x \cdot (h_p(x,t)p_t(x)) \log \frac{p_t(x)}{q_t(x)} \mathrm{d}x + \int \frac{p_t(x)}{q_t(x)} \nabla_x \cdot (h_p(x,t)p_t(x)) \mathrm{d}x \\
&= -\int p_t(x)[h_p(x,t) - h_q(x,t)]^\top [\nabla_x \log p_t(x) - \nabla_x \log q_t(x)] \mathrm{d}x \\
&= -\frac{1}{2}g^2(t) \int p_t(x) \|\nabla_x \log p_t(x) - \nabla_x \log q_t(x)\|_2^2 \mathrm{d}x \\
&= -\frac{1}{2}g^2(t)D_F(p_t||q_t)
\end{aligned}
\tag{S22}
$$

where $D_F(p_t||q_t)$ is the Fisher divergence. Considering that $g^2(t) = \beta(t) > 0$, and the Fisher divergence $D_F(p_t||q_t) \geq 0$ and the equality is established only if $p_t = q_t$, as a result, we have Eq S19, where the equality is established only if $p_t = q_t$.

## E  EXPERIMENTS ON THE QUALCOMM KEYWORD SPEECH DATASET

In addition to the commonly used SC09, for a more comprehensive consideration, we also conduct experiments on the Qualcomm Keyword Speech Dataset (Kim et al., 2019), denoted as QKW in the following. QKW consists of 4270 utterances belonging to four classes, with variable durations from 0.48s to 1.92s. We split them into a training set (3770 utterances), a validation set (400 utterances), and a test set (100 utterances). To handle the variable-sized input, we train an Attention Recurrent Convolutional Network (Shan et al., 2018) and save the checkpoint with the highest accuracy on the validation set. Then we finetuned the DiffWave model on QKW for 50,000 steps, with $lr = 2e - 4$ and $batch\_size\_per\_gpu = 2$ for 3 GPU. The results under $\mathcal{L}_\infty$ PGD$_{10}$ with $\epsilon = 0.002$ are shown in Table B. We can observe that AudioPure can still achieve non-trivial robustness and handle the audio with variable time duration well.

Table B: We apply AudioPure to the Qualcomm Keyword Speech Dataset. The diffusion steps $n^\star$ is set to 2.

| Defense | Clean | Robust |
|---------|-------|--------|
| None | 100 | 0 |
| AudioPure | 91 | 61 |

## F  FINE-TUNING ON ADVERSARIAL EXAMPLES

AudioPure takes advantage of pretrained diffusion models. We wonder whether the purification performance will be improved if fine-tuned on adversarial examples. And we further fine-tune the DiffWave model by augmenting self-supervised perturbation (SSP) (Naseer et al., 2020). Specifically, we use STFT (rescaling to the Mel-scale) as our feature extractor and maximize the following objective to generate perturbed examples:

$$
\arg\max_{x'} \Delta(x, x') = \|STFT(x), STFT(x')\|_\infty, \quad s.t. \|x - x'\|_\infty
\tag{S23}
$$

where $x$ is the clean example and $x'$ is the perturbed example. We then use gradient descent to optimize the perturbed example by:

$$
x'_{t+1} = clip(x'_t + \alpha \cdot sign(\nabla_x \Delta(x, x'_t)), x - \epsilon, x + \epsilon),
\tag{S24}
$$

for $t = 1, \ldots, T$. Here we use $T = 100$, $\epsilon = 0.002$ and $\alpha = 0.0004$. Next, we fine-tune the pretrained DiffWave model on the SSP examples, minimizing the following loss:

$$
\mathcal{L}_{tuning} = \mathcal{L}_{audio} + \lambda \mathcal{L}_{feat},
\tag{S25}
$$

where

$$\mathcal{L}_{audio} = MSE(x, \textbf{Purifier}(x'_t, n^\star)), \tag{S26}$$

$$\mathcal{L}_{feat} = MSE(STFT(x), STFT(\textbf{Purifier}(x'_t, n^\star))). \tag{S27}$$

We choose $\lambda = 0.1$ and use SGD to optimize $\mathcal{L}_{tuning}$, setting the learning rate to $1e-5$. The results are shown in Table C. As a result, it does not improve the performance of AudioPure (with $n^\star = 3$) under $\mathcal{L}_\infty$ PGD$_{10}$ and PGD$_{70}$ with $\epsilon = 0.002$. These results further verify the effectiveness of using pretrained models.

Table C: We fine-tune the pretrained DiffWave model on adversarial examples generated by SSP. After fine-tuning, the performance is not improved.

| Defense | Clean | PGD$_{10}$ | PGD$_{70}$ |
|---|---|---|---|
| None | 100 | 3 | 1 |
| AudioPure | 97 | 89 | 84 |
| SSP-Tuned AudioPure | 97 | 89 | 82 |

## G    COMPARISON WITH OTHER DENOISER-BASED DEFENSE

We compare AudioPure with DefenseGAN (Samangouei et al., 2018) and Joint Adversarial Fine-tuning (Joshi et al., 2022). For DefenseGAN, which is originally designed to defend against adversarial images by finding the optimal noise that generates the most similar image to the adversarial counterpart, we adopt it to the audio domain, choosing WaveGAN (Donahue et al., 2018) as the GAN model in this pipeline. We train a WaveGAN on the SC09 dataset for 100 epochs, using the Adam optimizer with $lr = 1e-3$, $\beta_1 = 0.5$, and $\beta_2 = 0.9$. For Joint Adversarial Fine-tuning, we follow the setting of Joshi et al. (2022), using a Conv-TasNet (Luo & Mesgarani, 2019) as the denoiser. And like Joshi et al. (2022), we craft an offline adversarial SC09 dataset against the pretrained classifier by using L-inf PGD-100 attacks with $\epsilon = 0.002$ (denoted as OffAdv-SC09). Then we train a Conv-TasNet model on OffAdv-SC09 for 30 epochs to get the pretrained denoiser. We denote the defense using the pretrained Conv-TasNet as CTN Baseline. Based on the adversarial examples generated by attacking the whole acoustic system, we only update the Conv-TasNet denoiser while keeping the classifier frozen, and denote this method as CTN Adv-Finetune-Joint-frozen. During the adversarial tuning, we use $\mathcal{L}_\infty$ PGD$_{10}$ attack with $\epsilon = 0.002$. After tuning for 1000 steps with $batch\_size = 20$, we calculate the clean and robust accuracy (under $\mathcal{L}_\infty$ PGD$_{10}$ and PGD$_{70}$ with $\epsilon = 0.002$) on the same test used in our paper.

We report the results in Table D. We find that DefenseGAN based on WaveGAN cannot work well in the audio domain. It shows the impact of domain differences with respect to the final results and verifies the importance of our pipeline design. Besides, the Conv-TasNet denoiser is less effective than diffusion models against adaptive attacks, even after fine-tuning.

Table D: We compare AudioPure with different denoiser-based defenses. DiffWave is proven to be a more effective purifier.

| Defense | Clean | PGD$_{10}$ | PGD$_{70}$ |
|---|---|---|---|
| None | **100** | 3 | 1 |
| AudioPure | 97 | **89** | **84** |
| DefenseGAN | 8 | 0 | 0 |
| CTN Baseline | 98 | 13 | 1 |
| CTN Adv-Finetune-Joint-frozen | 90 | 52 | 41 |

## H  COMPARISON WITH THE REGULARIZATION-BASED DEFENSE

Gu & Rigazio (2014); Hoffman et al. (2019) introduce the input-output Jacobian matrix of the network as a regularization term in the optimization objective, formulated as

$$\mathcal{L}_{reg} = \sum_i \left( \mathcal{L}(x_i, y_i) + \lambda \| \frac{\partial f(x_i)}{\partial x_i} \|_F \right), \tag{S28}$$

where $x_i \in \mathbb{R}^d$ is the input data, $y_i \in \mathbb{R}^n$ is the label, $\mathcal{L} : \mathbb{R}^d \times \mathbb{R}^n \to \mathbb{R}$ is the original loss function, and $f : \mathbb{R}^d \to \mathbb{R}^n$ is the neural network. By minimizing the Frobenius norm of the Jacobian matrix, the adversarial robustness of the network will be improved. For a more comprehensive study, we also compare AudioPure with this regularization-based method, using different $\lambda$. The results are shown in Table E, where we denote the regularization-based defense as *Jacobian-Reg*.

Table E: We compare AudioPure with the regularization-based defense, using different $\lambda$.

| Defense | Clean | $PGD_{10}$ | $PGD_{70}$ |
|---|---|---|---|
| None | **100** | 3 | 1 |
| AudioPure | 97 | **89** | **84** |
| Jacobian-Reg ($\lambda$=1e-8) | 45 | 9 | 5 |
| Jacobian-Reg ($\lambda$=1e-9) | 84 | 27 | 15 |
| Jacobian-Reg ($\lambda$=1e-10) | 91 | 31 | 18 |
| Jacobian-Reg ($\lambda$=1e-11) | 96 | 19 | 4 |

## I  EXPERIMENTS ON LARGER ATTACK BUDGETS.

Besides the results of different $\epsilon$ in Table 2, we conduct additional experiments to explore the potential of the diffusion model for purification. We select $\epsilon = \{0.01, 0.02, 0.03, 0.04, 0.05\}$, and set the diffusion steps $n^\star$. The results are shown in Table F. We find that our method still achieves 42% accuracy at $\epsilon = 0.03$, which brings significant distortions to audio. Our method keeps the ability to purify adversarial perturbations until $\epsilon = 0.05$. We also visualize the audio waveforms under attacks with different $\epsilon$, illustrated in Figure B. It is easy to observe significant noise in them.

Table F: We explore the potential of DiffWave under larger attack budgets. The diffusion steps $n^\star$ is set to 5.

| Attack Budget | $\epsilon = 0.01$ | $\epsilon = 0.02$ | $\epsilon = 0.03$ | $\epsilon = 0.04$ | $\epsilon = 0.05$ |
|---|---|---|---|---|---|
| Robust Acc. | 82 | 67 | 42 | 14 | 0 |

## J  ADDITIONAL INFERENCE TIME COST.

Due to the introduction of diffusion models, AudioPure will bring additional time cost during inference. As shown in Table G, we compute the time cost per audio, averaged on 100 examples and the time duration for each example is around one second. We evaluate it on an NVIDIA RTX 3090 GPU with Intel® Core™ i9-10920X CPU @ 3.50GHz and 64 GB RAM.

Table G: The inference time cost when using different diffusion steps $n^\star$.

| Diffusion Steps | $n^\star = 0$ | $n^\star = 1$ | $n^\star = 2$ | $n^\star = 3$ | $n^\star = 5$ | $n^\star = 7$ | $n^\star = 10$ |
|---|---|---|---|---|---|---|---|
| Time Cost (s) | 0.0967 | 0.5522 | 0.7876 | 1.0162 | 1.4795 | 2.0125 | 2.6839 |

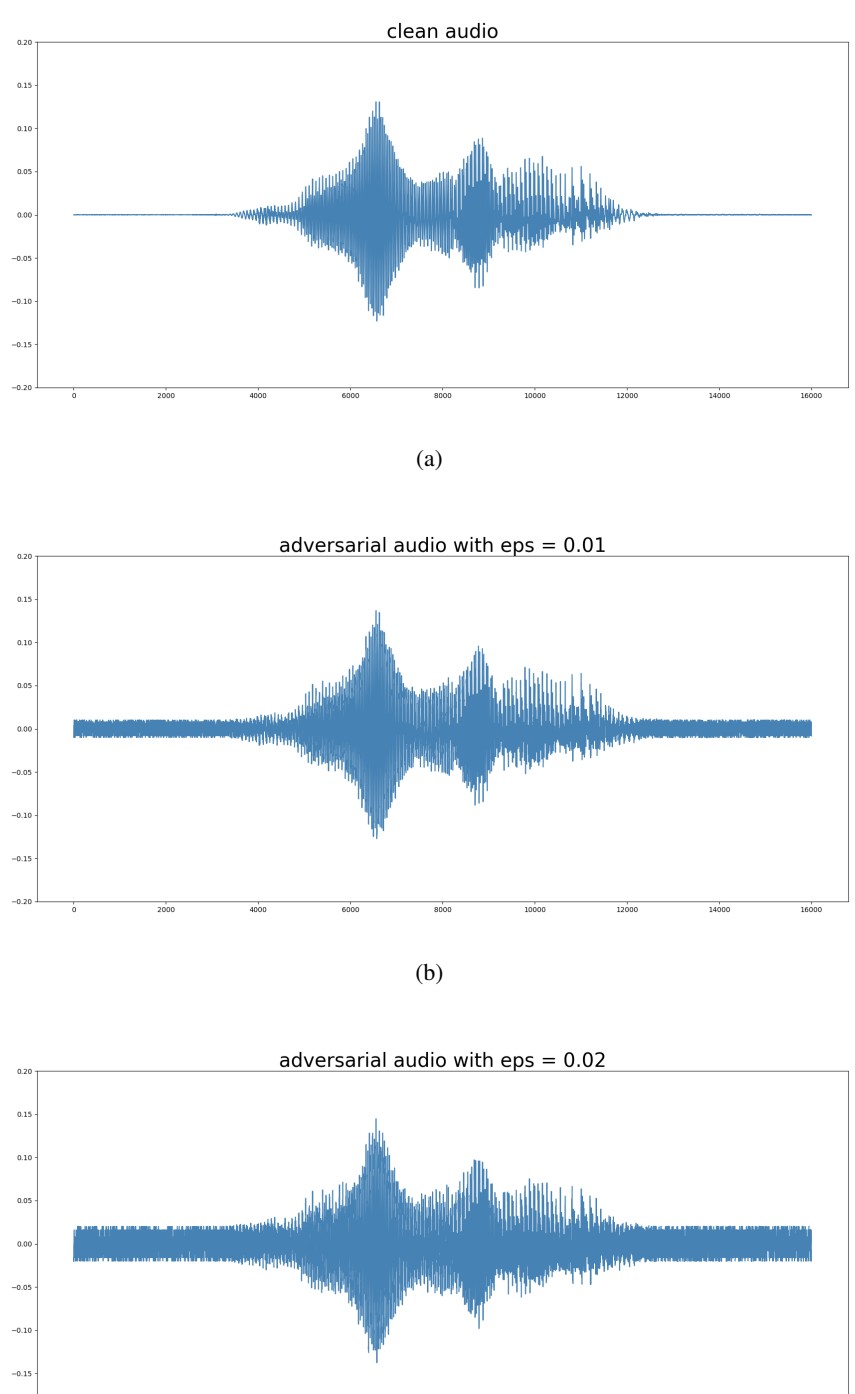

Figure B: Visualizations of the clean audio and adversarial audio with different attack budgets.

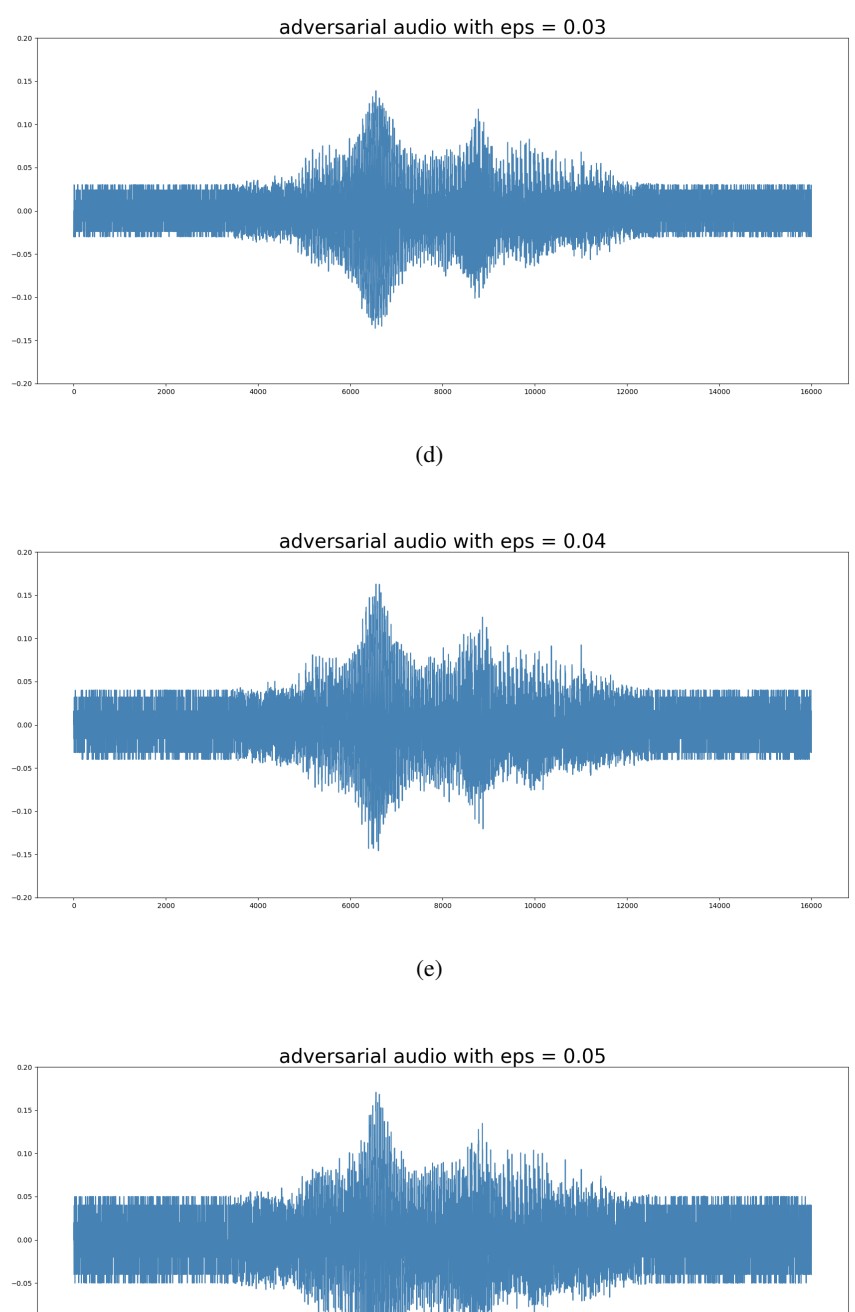

(d)

(e)

(f)

Figure B: Visualizations of the clean audio and adversarial audio with different attack budgets.

