# OpenReview forum: "Defending against Adversarial Audio  via Diffusion Model"
_ICLR.cc/2023/Conference — ICLR 2023 poster_

### Official Review · Reviewer_phaA · 2022-10-23

**Confidence:** 4
**Correctness:** 3
**Technical Novelty And Significance:** 3
**Empirical Novelty And Significance:** 3
**Recommendation:** 6

**Clarity, Quality, Novelty And Reproducibility:**

This paper employs the diffusion model for the protection of acoustic systems. The novelty is not sufficient, so I wonder if better adaptive adversarial finetune strategy can be achieved for the diffusion model. The clarity is basically correct.

**Strength And Weaknesses:**

Strength:
This is the first work using the diffusion model to achieve the robustness of the acoustic systems, and the results demonstrate the effectiveness of the proposed method.

Weakness:
1. The experiments are only conducted on one dataset, and more datasets should be involved to show the generalization of the proposed strategy.

2. As shown in Table 2, there are different optimized diffusion steps faced with different attack budgets. And how to decide the optimal value for an input audio with unknown budgets?

3. If the diffusion model is finetuned according to the input adversarial audio samples, like the strategy of [A], will be performance be improved?

[A] A self-supervised approach for adversarial robustness, CVPR2020

**Summary Of The Paper:**

This paper proposes an adversarial purification-based defense strategy to defend against adversarial audio examples. This method is called AudioPure which is achieved by the off-the-shelf diffusion model's generation. AudioPure adds noise to the adversarial audio and then utilizes the reverse sampling step to purify the noisy audio. AudioPure is employed as a plug-and-play module for any pretrained classifier. The experiments are conducted on representative datasets to show the robustness achieved by AudioPure under white-box and black-box attacks.

**Summary Of The Review:**

This paper proposes a diffusion-based strategy for the adversarial defense of the acoustic systems. And my main concerns are the novelty and the experimental details.

---

> ### Author Response · Authors · 2022-11-19
> **Response to Reviewer phaA**
>
> Thanks for your valuable comments. We address your concerns below:
>
> >Q1. The experiments are only conducted on one dataset, and more datasets should be involved to show the generalization of the proposed strategy.
>
> A1. Thanks for your suggestion. We have added experiments on another keyword spotting task using Qualcomm Keyword Speech Dataset[1], denoted as QKW in the following. QKW consists of 4270 utterances belonging to four classes, with variable durations from 0.48s to 1.92s. We split them into a training set (3770 utterances), a validation set (400 utterances), and a test set (100 utterances). To handle the variable-sized input, we train an Attention Recurrent Convolutional Network[2] on the training set using the code from [3], and save the checkpoint with the highest accuracy on the validation set. Then we finetuned the DiffWave model on QKW for 50,000 steps, with $lr=2e-4$ and batch_size_per_gpu=2 for 3 GPU.
>
> Due to the time limitation, we only evaluate $n=2$. The results are shown in the following. We can observe that AudioPure can still achieve non-trivial robustness and handle the audio with variable time duration well.
>
> |Defense|Clean|Robust|
> |---|---|---|
> |None|100|0|
> |AudioPure (QKW finetuned)|91|61|
>
> >Q2. As shown in Table 2, there are different optimized diffusion steps faced with different attack budgets. And how to decide the optimal value for input audio with unknown budgets?
>
> A2. Thanks for pointing it out. From Table 2, we can find that $n=5$ can consistently provide non-trivial adversarial robustness. Thus, we can use $n=5$ as the empirical threshold. We believe how to dynamically choose the $n$ for the different inputs is a valuable and open question. Here, we provide one potential solution: we leverage a pretrained speech denoising model to denoise the input audio, then we calculate the SNR between the denoised audio and original audio. If the perturbation is large, it may result in a large SNR. Thus, we train a model to map SNR to the output time step $n$.
>
> >Q3. If the diffusion model is finetuned according to the input adversarial audio samples, like the strategy of [A], will performance be improved?
> Reference: [A] A self-supervised approach for adversarial robustness, CVPR2020
>
> A3. Thanks for your insightful idea. We have conducted experiments to explore it further.
>
>
> we further augment self-supervised Perturbation (SSP) [1] to finetune the DiffWave model.
>
>
> Specifically, we use STFT (rescaling to the Mel-scale) as our feature extractor and maximize the following objective to generate perturbed examples:
> $$ max \;\Delta(x, x^\prime)=\Vert STFT(x), STFT(x^\prime) \Vert_\infty, \quad s.t. \Vert x-x^\prime \Vert_\infty \leq \epsilon ,$$
> where $x$ is the clean example and $x^\prime$ is the perturbed example. We then use gradient descent to optimize the perturbed example by:
> $$ x^\prime_{t+1} = clip(x^\prime_{t} + \alpha \cdot sign(\nabla_x \Delta(x, x^\prime_{t})), x-\epsilon, x+\epsilon)$$
> for $t = 1, \dots, T$. Here we use $T=100$, $\epsilon=0.002$ and $\alpha=0.0004$.
> Next, we finetune the pretrained DiffWave model on the SSP examples, minimizing the following loss:
> $$ L = L_{audio} + \lambda L_{feat}, $$
> where
> $$ L_{audio} = MSE(x, \mathbf{Purifier}(x^\prime_{t}, n^\star)), $$
> $$ L_{feat} = MSE(STFT(x), STFT(\mathbf{Purifier}(x^\prime_{t}, n^\star)) .$$
>
> We use SGD to optimize $L$, setting the learning rate to 1e-5. The results are shown in the following table.
> As a result, it only slightly improved the performance of AudioPure. These results further verify the effectiveness of using the pretrained models.
>
> |Defense|Clean|Robust|
> |---|---|---|
> |None|100|3|
> |AudioPure|96|84|
> |SSP-Tuned AudioPure|96|85|
>
> Reference:
>
> [1] https://developer.qualcomm.com/project/keyword-speech-dataset
>
> [2] Shan, Changhao, et al. "Attention-based End-to-End Models for Small-Footprint Keyword Spotting." Proc. Interspeech 2018 (2018): 2037-2041.
>
> [3] https://github.com/Kirili4ik/kws-attention-pytorch
>
> [4] Naseer, Muzammal, et al. "A self-supervised approach for adversarial robustness." Proceedings of the IEEE/CVF Conference on Computer Vision and Pattern Recognition. 2020.

---

> ### Author Response · Authors · 2022-11-28
> **Looking forward to further discussion**
>
> Dear Reviewer,
>
> We are grateful for your valuable comments. In our response, we have included additional experiments on a different dataset, and explained the optimal choice of diffusion steps $n$. Moreover, we have finetuned the diffusion model according to adversarial examples, and the results are included. We hope that your concerns have been addressed.
>
> As you might know that we are inching closer to the reviewer final recommendation deadline. Thus we would really appreciate it if you can provide us the feedback on our detailed rebuttal. We have incorporated all the changes in our revised manuscript for your kind consideration.
>
> If you feel happy with our response, please consider updating your score. Feel free to reach us out in case you need any clarification.
>
> Paper4179 Authors

---

> ### Author Response · Authors · 2022-12-02
> **Looking forward to further discussion**
>
> Dear Reviewer,
>
> As you might notice, the due of the discussion stage is approaching, and your recommendation is vital to our work. We would appreciate it if you could provide feedback on our detailed rebuttal. If you are satisfied with our response, please kindly reconsider your score. And please feel free to further discuss with us if there is any other question.
>
> Sincerely,
>
> Paper4179 Authors

---

> ### Author Response · Authors · 2022-12-08
> **Have we addressed your concerns?**
>
> Dear Reviewer,
>
> The discussion stage is nearing its end. However, we are unsure yet whether your concerns have been addressed in our response, and we sincerely look forward to your vital feedback. If you are satisfied with our response, please kindly reconsider your score. And if there are any other new questions, please feel free to discuss them further with us.
>
> Best regards,
>
> Paper4179 Authors

---

### Official Review · Reviewer_NJPC · 2022-10-25

**Confidence:** 4
**Correctness:** 3
**Technical Novelty And Significance:** 2
**Empirical Novelty And Significance:** 3
**Recommendation:** 6

**Clarity, Quality, Novelty And Reproducibility:**

The authors state they will release the code and models. Details seem to be sufficient that, once such release is public, results can be reproduced.

**Details Of Ethics Concerns:**

I don't see any particular ethic issue, beyond potentially access to the technology due to increase (train) inference cost.
In general given the strong results provided in term of resilience to attacks, such "cost" is very likely warranted.

**Strength And Weaknesses:**

The paper is very extensive, both in term of motivation, background, details of the experiments.
The performance especially for white-box / adaptive attacks is rather impressive. This seems to be a trend with the latest diffusion-based adversarial robustness methods.
Actual quantification of inference time increase / cpu requirements would be good to have to complete the picture in term of practical applicability of the method proposed.
One area about adversarial robustness is missing, the regularization based defenses, with seminal paper: https://arxiv.org/pdf/1412.5068.pdf


**Summary Of The Paper:**

The paper introduces a DiffPure inspired method (Diffusion models for adversarial robustness) in the context of speech-reco acoustic models. The adaptation of DiffPure, called AudioPure, operates in time domain with a diffusion process before the stft/mel feature extraction. The paper focuses on measuring the performance of different attack strategies, including white-box attacks where the diffusion approach really shines.

**Summary Of The Review:**

This is a good research, in term of adapting a very promising approach of adversarial robustness (DiffPure) from image to speech.
Because of that this work is not completely new, but more like a solid port of an existing strong baseline from a different modality to the speech domain. That said, the details and the strength of the results may still warrant relatively broad interest for the work, especially after few tweaks and improvements.

---

> ### Author Response · Authors · 2022-11-19
> **Response to Reviewer NJPC (Part 1)**
>
> Thanks for your valuable comments. We address your concerns below:
>
> >Q1. Actual quantification of inference time increase / CPU requirements would be good to have to complete the picture in terms of the practical applicability of the method proposed.
>
> A1. We compute the time cost per audio, averaged on 100 examples and the time duration for each example is around 1s on average. We evaluate it on an NVIDIA RTX 3090 GPU with Intel(R) Core(TM) i9-10920X CPU @ 3.50GHz and 64 GB Mem.
>
> |Diffusion Steps|n=0|n=1|n=2|n=3|n=5|n=7|n=10|
> |---|---|---|---|---|---|---|---|
> |Time Cost(s)|0.0967|0.5522|0.7876|1.0162|1.4795|2.0125|2.6839|
>
> Table 2 in the paper shows that when n=3, it can still achieve non-trivial robust accuracy while the time complexity is only 1.0162s on average, which is applicable for most real-time applications since the original audio length is around 1s.
>
> >Q2. One area about adversarial robustness is missing, the regularization-based defenses, with seminal paper: https://arxiv.org/pdf/1412.5068.pdf
>
> A2. Thanks for your suggestion. We conduct additional experiments with this method. We use this GitHub repository[1] to implement Jacobian regularization. The following table shows the results when using different regularization weight $\lambda$ (equation 4 in [2]):
>
> |Defense|Clean|Robust|
> |---|---|---|
> |None|100|0|
> |AudioPure|96|84|
> |Jacobian-Reg ($\lambda$=1e-8)|45|9|
> |Jacobian-Reg ($\lambda$=1e-9)|84|27|
> |Jacobian-Reg ($\lambda$=1e-10)|91|31|
> |Jacobian-Reg ($\lambda$=1e-11)|96|19|
>
> We can observe that our method achieves higher performance than the regularization-based defense. We will add these results to our main paper.
>
> Reference:
>
> [1] https://github.com/facebookresearch/jacobian_regularizer
>
> [2] Gu, Shixiang, and Luca Rigazio. "Towards deep neural network architectures robust to adversarial examples." arXiv preprint arXiv:1412.5068 (2014).

---

> ### Author Response · Authors · 2022-11-19
> **Response to Reviewer NJPC (Part 2)**
>
> >Q3. This is a good research, in terms of adapting a very promising approach of adversarial robustness (DiffPure) from image to speech. Because of that this work is not completely new, but more like a solid port of an existing strong baseline from a different modality to the speech domain. That said, the details and the strength of the results may still warrant relatively broad interest for the work, especially after a few tweaks and improvements.
>
> A3: Thanks for summarization. Although AudioPure is an adaptation of DiffPure from the image domain, such adaptation is non-trivial. Audio signals have some unique properties. For instance, there are different choices of audio representations, including raw waveforms and various types of time-frequency representations (e.g., Mel spectrogram, MFCC). When designing an acoustic system, some particular audio representations may be selected as the target features, and defenses that work well on some features may perform poorly on other features. In this paper, if we directly treat the 2-D time-frequency representations (i.e., spectrogram) as images and then directly apply the successful DiffPure from the image domain for the spectrogram, it achieves a lower robustness demonstrated in this work.
>
> Additionally, the acoustic system can take audio with variable time duration as the input, while the underlying diffusion model within
> DiffPure can only handle inputs with fixed width and height. We conduct additional experiments on keyword sporting tasks using Qualcomm Keyword Speech Dataset [1], denoted as QKW in the following. QKW consists of 4270 utterances belonging to four classes, with variable durations from 0.48s to 1.92s. We split them into a training set (3770 utterances), a validation set (400 utterances), and a test set (100 utterances). To handle the variable-sized input, we train an Attention Recurrent Convolutional Network[2] on the training set using the code from [3], and save the checkpoint with the highest accuracy on the validation set. Then we finetuned the DiffWave model on QKW for 50,000 steps, with $lr=2e-4$ and batch_size_per_gpu=2 for 3 GPU.
> Due to the time limitation, we only evaluate $n=2$. The results are shown in the following. We can observe that AudioPure can still achieve non-trivial robustness and handle the audio with variable time duration well.
>
>
> |Defense|Clean|Robust|
> |---|---|---|
> |None|100|0|
> |AudioPure|91|61|
>
>
> Last, we also highlight that the methods that work in image domain may not work in the audio domain. For instance, we conduct additional experiments by adapting DefenseGAN to the audio domain. We choose WaveGAN[4] as the GAN model. We refer to the PyTorch implementation from [5] and train a WaveGAN on the SC09 dataset for 100 epochs, using Adam with $lr=0.001$, $\beta_1=0.5$ and $\beta_2=0.9$. The results are shown in the following. We find DefenseGAN based on WaveGAN cannot work well in the audio domain. It shows the impact of domain differences with respect to the final results and verifies the importance of our pipeline design.
>
> |Defense|Clean|Robust|
> |---|---|---|
> |None|100|3|
> |AudioPure|96|84|
> |DefenseGAN(WaveGAN)|8|0|
>
> Reference:
>
> [1] https://developer.qualcomm.com/project/keyword-speech-dataset
>
> [2] Shan, Changhao, et al. "Attention-based End-to-End Models for Small-Footprint Keyword Spotting." Proc. Interspeech 2018 (2018): 2037-2041.
>
> [3] https://github.com/Kirili4ik/kws-attention-pytorch
>
> [4] Samangouei, Pouya, Maya Kabkab, and Rama Chellappa. "Defense-gan: Protecting classifiers against adversarial attacks using generative models." arXiv preprint arXiv:1805.06605 (2018).
>
> [5] https://github.com/lukysummer/WaveGAN-Speech-Synthesis

---

> ### Author Response · Authors · 2022-11-28
> **Looking forward to further discussion**
>
> Dear Reviewer,
>
> We really appreciate your valuable comments. In our response, we have added the inference time cost when using different diffusion steps $n$, and included additional experiments on regularization-based defense. Besides, we have explained the non-triviality of the adaptation from the image domain to the audio domain. We hope that your concerns have been addressed.
>
> As you might know that we are inching closer to the reviewer final recommendation deadline. Thus we would really appreciate it if you can provide us the feedback on our detailed rebuttal. We have incorporated all the changes in our revised manuscript for your kind consideration.
>
> If you feel happy with our response, please consider updating your score. Feel free to reach us out in case you need any clarification.
>
> Paper4179 Authors

---

> ### Author Response · Authors · 2022-12-02
> **Looking forward to further discussion**
>
> Dear Reviewer,
>
> As you might notice, the due of the discussion stage is approaching, and your recommendation is vital to our work. We would appreciate it if you could provide feedback on our detailed rebuttal. If you are satisfied with our response, please kindly reconsider your score. And please feel free to further discuss with us if there is any other question.
>
> Sincerely,
>
> Paper4179 Authors

---

> ### Author Response · Authors · 2022-12-08
> **Have we addressed your concerns?**
>
> Dear Reviewer,
>
> The discussion stage is nearing its end. However, we are unsure yet whether your concerns have been addressed in our response, and we sincerely look forward to your vital feedback. If you are satisfied with our response, please kindly reconsider your score. And if there are any other new questions, please feel free to discuss them further with us.
>
> Best regards,
>
> Paper4179 Authors

---

> ### Author Response · Authors · 2022-12-12
> **Any pending questions?**
>
> Dear reviewer,
>
> We would appreciate it if you could provide feedback on our detailed rebuttal. If you are satisfied with our response, please kindly consider increasing your score. And please feel free to let us know if there is any other question.

---

> > ### Comment · Reviewer_NJPC · 2022-12-13
> > **Author feedback accepted**
> >
> > Dear Authors sorry for the delay (traveling),
> >
> > your extensive responses to all reviewers questions has indeed been highly appreciated. For this specific review your results on the computation time are very useful. As stated before your proposed method is showing strong results.
> >
> > Because of that and your creativity in addressing reviewer questions I would like to upgrade my review from 5 to 6 (above acceptance)
> >
> > Thank you.

---

> > > ### Author Response · Authors · 2022-12-13
> > > **Thank you for the reply**
> > >
> > > Dear Reviewer,
> > >
> > > We really appreciate your reply. We are glad that our responses have addressed your concerns and your would like to raise the score from 5 to 6.
> > >
> > > By the way, the recommendation score in the review seems unchanged. Could you nicely upadate it? Many thanks!
> > >
> > > Best,
> > >
> > > Authors

---

### Official Review · Reviewer_fScq · 2022-10-27

**Confidence:** 3
**Clarity, Quality, Novelty And Reproducibility:** 1. Clear enough. Some sections can be…
**Correctness:** 3
**Technical Novelty And Significance:** 3
**Empirical Novelty And Significance:** 3
**Recommendation:** 8

**Strength And Weaknesses:**

Strengths:
1. Excellent idea which is well described
2. The empirical evaluations are also well described and insightful.

Weakness:
1. the technical sections in Section 3 are still a difficult read and the presentation can be improved.
2. It is not clear how the diffusion process to a certain extent removes the adversarial perturbation. I thought this would be the most interesting question to be addressed by this paper but I was not able to find a crisp answer to this question.

**Summary Of The Paper:**

The authors address the problem of defending against adversarial audio attacks using audio purification based on recently popular diffusion models for recovering clean audio. Overall the idea is quite interesting and the experimental evaluation shows strong performance of the proposed idea.

**Summary Of The Review:**

Overall I really liked the idea of using the diffusion model to re-generate clean audio from adversarial audio in order to defend against adversarial attacks. The paper is well written with insightful experiments.

---

> ### Author Response · Authors · 2022-11-19
> **Response to Reviewer fScq**
>
> We are grateful for your valuable comments and your appreciation of our idea. We address your concerns below:
>
> >Q1. the technical sections in Section 3 are still a difficult read and the presentation can be improved.
>
> A1. We promise to improve the presentation quality of Section 3 to make it more readable.
>
> >Q2. It is not clear how the diffusion process to a certain extent removes the adversarial perturbation. I thought this would be the most interesting question to be addressed by this paper but I was not able to find a crisp answer to this question.
>
> A2. Thanks for your great question. In table 2, we have the results with different $\epsilon$. Here, we conduct additional experiments to explore the extent of the diffusion model for purification. We selet $\epsilon=\{0.01,0.02,0.03,0.04,0.05\}$. Due to the time limitation, for all $\epsilon$, we use the same $t=5$. The results are shown in the following table. We find that our method still achieves 42% accuracy at $\epsilon=0.03$. When $\epsilon=0.05$, our method does not work.
>
> We also visualize the audio with these $\epsilon$ . We can observe significant noise on these audios. We conduct a small-scale human study with 5 people due to the time limitation. We asked the participants to identify whether they could hear the apparent noise. 100% of users can identify the noise even with $\epsilon=0.01$.
>
>
> |Attack Budget|$\epsilon=0.01$|$\epsilon=0.02$|$\epsilon=0.03$|$\epsilon=0.04$|$\epsilon=0.05$|
> |---|---|---|---|---|---|
> | Robust Acc. | 82 | 67 | 42 | 14 | 0 |
>
> The visualization of the clean audio and adversarial audio can be found in following anonymous links:
>
> clean audio: https://i.imgur.com/DzBQ9s7.png
>
> adversarial audio ($\epsilon = 0.01$): https://i.imgur.com/hLexFVP.png
>
> adversarial audio ($\epsilon = 0.02$): https://i.imgur.com/aCfOLOs.png
>
> adversarial audio ($\epsilon = 0.03$): https://i.imgur.com/pe5dSPd.png
>
> adversarial audio ($\epsilon = 0.04$): https://i.imgur.com/Eo9Hj4K.png
>
> adversarial audio ($\epsilon = 0.05$): https://i.imgur.com/iCx8ajQ.png

---

> > ### Comment · Reviewer_fScq · 2022-12-02
> > **Thank you for your careful responses.**
> >
> > The authors have done a good job of addressing some of the reviewers concern. I am happy with thier response and would like to stick to my original assessment.

---

> > > ### Author Response · Authors · 2022-12-03
> > > **Thank you**
> > >
> > > We sincerely appreciate your approbation of our response. If you have any other new questions, please feel free to discuss them with us.

---

> ### Author Response · Authors · 2022-11-28
> **Looking forward to further discussion**
>
> Dear Reviewer,
>
> We are grateful for your valuable comments. In our response, we have included additional experiments on different attack budgets to investigate how the diffusion process to a certain extent removes the adversarial perturbation. We hope that your concerns have been addressed.
>
> As you might know that we are closer to the reviewer final recommendation deadline. Thus we would really appreciate it if you can provide us the further feedback on our detailed rebuttal. We have incorporated all the changes in our revised manuscript for your kind consideration.
>
> If you have other questions or need any clarification, please feel free to reach us out.
>
> Paper4179 Authors

---

### Official Review · Reviewer_6825 · 2022-10-27

**Confidence:** 4
**Correctness:** 3
**Technical Novelty And Significance:** 3
**Empirical Novelty And Significance:** 3
**Recommendation:** 8

**Clarity, Quality, Novelty And Reproducibility:**

Paper is very clear. Although, continuous-time diffusion equations are hard to understand for readers without background.
Novelty is low as plug-and-play scheme of a model is too simple. Perhaps using multiple diffusion models offer interesting solutions.
Reproducibility is not an issue, in my opinion.

**Strength And Weaknesses:**

Strengths:
1. Providing code is encouraging.
2. Breath of experiments is great.
3. Paper is good step in promoting diffusion models for adversarial defense.

Limitations:
1. Using existing pre-trained models as defense may not be enough to satisfy novelty requirements. The training of DiffWave and DiffSpec is not modified to account for adversarial noise.
2. Comparison with four-year old Adversarial training is not enough. DefenseGAN and denoiser defense are close to state-of-the-art defenses. Examples of such works:

Samangouei, Pouya, Maya Kabkab, and Rama Chellappa. "Defense-gan: Protecting classifiers against adversarial attacks using generative models." arXiv preprint arXiv:1805.06605 (2018).

Joshi, Sonal, et al. "Defense against Adversarial Attacks on Hybrid Speech Recognition using Joint Adversarial Fine-tuning with Denoiser." arXiv preprint arXiv:2204.03851 (2022).

3. "Limitation" section is hastily written. It may be removed.
4. I could not understand the intuition behind the proposal. Why should diffusion models work for cleaning adversarial signals?

**Summary Of The Paper:**

This work proposes a technique called AudioPure where pre-trained diffusion-based speech denoisers are used for adversarial defense. The diffusion model is intended to be used in forward and backward steps to arrive from adversarial signal to purified audio. The paper proposes to use DiffWave and DiffSpec to be able to operate in time and time-frequency domains. The theory behind SDE is thoroughly discussed. SDE modifications to get cleaner audio is very interesting.
The choice of end application is Speech Command Recognition. There are many experiment sub-sections. Mainly the proposed technique is compared with Adversatial training method. Tuning experiments include varying PGD steps for attack, EOT size, n*, choice of classifier,
audio features, and RS variance. In all experiments, proposed technique shows promise.

**Summary Of The Review:**

The choice of problem is good. The choice of model (diffusion) is also interesting. The breath of experiments is impressive.
Although as explained earlier, novelty is low. Investigating diffusion models specifically trained for cleaning adversarial audio should be prefered in revision of paper.
Also, stronger evaluation (higher eps) is needed while also comparing with DefenseGAN and denoiser defenses.

---

> ### Author Response · Authors · 2022-11-19
> **Response to Reviewer 6825 (Part 1)**
>
> Thanks for your valuable comments. We address your concerns below:
>
> >Q1. Using existing pre-trained models as defense may not be enough to satisfy novelty requirements. The training of DiffWave and DiffSpec is not modified to account for adversarial noise.
>
> A1. Thanks for pointing it out. We think using the existing off-the-shelf diffusion model as the defense method in a plug-and-play manner is the advantage of our method. In this way, we can easily deploy the off-the-shelf diffusion models and classifier to enhance the robustness without additional training.
>
> Additionally, as mentioned in the introduction, There are different choices of audio representations, including raw waveforms and various types of time-frequency representations (e.g., Mel spectrogram, MFCC). When designing an acoustic system, some particular audio
> representations may be selected as the target features, and defenses that work well on some features may perform poorly on others. Therefore, how to combine the diffusion models and classifier with different audio representations still need to be discovered and explored.
>
> On the other hand,  DiffSpec is not provided. Instead, we train the DiffSpec on the Mel-spectrogram data generated by SC09 audio.
>
> Based on reviewer's question, instead of using the pretrained diffusion model,  we further finetune the DiffWave model by augmenting self-supervised Perturbation (SSP) [1].
>
> Specifically, we use STFT (rescaling to the Mel-scale) as our feature extractor and maximize the following objective to generate perturbed examples:
> $$ max  \Delta(x, x^\prime)=\Vert STFT(x), STFT(x^\prime) \Vert_\infty, \quad s.t. \Vert x-x^\prime \Vert_\infty \leq \epsilon ,$$
> where $x$ is the clean example and $x^\prime$ is the perturbed example. We then use gradient descent to optimize the perturbed example by:
> $$ x^\prime_{t+1} = clip(x^\prime_{t} + \alpha \cdot sign(\nabla_x \Delta(x, x^\prime_{t})), x-\epsilon, x+\epsilon)$$
> for $t = 1, \dots, T$. Here we use $T=100$, $\epsilon=0.002$ and $\alpha=0.0004$.
> Next, we finetune the pretrained DiffWave model on the SSP examples, minimizing the following loss:
> $$ L = L_{audio} + \lambda L_{feat}, $$
> where
> $$ L_{audio} = MSE(x, \mathbf{Purifier}(x^\prime_{t}, n^\star)), $$
> $$ L_{feat} = MSE(STFT(x), STFT(\mathbf{Purifier}(x^\prime_{t}, n^\star)) .$$
>
> We use SGD to optimize $L$, setting the learning rate to 1e-5. The results are shown in the following table.
> As a result, it only slightly improved the performance of AudioPure. These results further verify the effectiveness of using pretrained models.
> |Defense|Clean|Robust|
> |---|---|---|
> |None|100|3|
> |AudioPure|96|84|
> |SSP-Tuned AudioPure|96|85|
>
> >Q2. Comparison with four-year-old Adversarial training is not enough. DefenseGAN and denoiser defense are close to state-of-the-art defenses. Examples of such works: [2][5].
>
> A2.  Thanks for the suggestion. We have added additional experiments on more recent works.
> (1) To adapt DefenseGAN[2] to audio settings, we choose WaveGAN[3] as the GAN model. We refer to the PyTorch implementation from [4] and train a WaveGAN on the SC09 dataset for 100 epochs, using Adam with $lr=0.001$, $\beta_1=0.5$ and $\beta_2=0.9$. The results are shown in the following. We find that DefenseGAN based on WaveGAN cannot work well in the audio domain. It shows the impact of domain differences with respect to the final results and verifies the importance of our pipeline design.
>
> |Defense|Clean|Robust|
> |---|---|---|
> |None|100|3|
> |AudioPure|96|84|
> |DefenseGAN(WaveGAN)|8|0|
>
> (2) For "denoiser defense" (i.e., the joint adversarial finetuning [5]), we follow the same methods in [5] and conduct additional experiments, including:
> * Baseline: it uses  the Conv-TasNet[6] denoiser pretrained on offline adversarial data and  pretrained ResNeXt classifier;
> * Adv-Finetune-Joint-Frozen: it uses pretrained Conv-TasNet denoiser and pretrained ResNeXt classifier, followed by adversarially training the denoiser.
>
> We refer to the code from [7] to build the Conv-TasNet denoiser.
>
> We craft an offline adversarial SC09 dataset against the pretrained classifier by using L-inf PGD-100 attacks with $\epsilon=0.002$ (denoted as OffAdv-SC09). Then we train a Conv-TasNet model on OffAdv-SC09 for 30 epochs to get the pretrained denoiser. We denote the defense using the pretrained Conv-TasNet as Denoiser Baseline.
> During the adversarial tuning, we use L-inf PGD-10 attack with $\epsilon=0.002$. After tuning for 1000 steps with batch_size = 20, we calculate the clean and robust accuracy (also under L-inf PGD-10 attack with $\epsilon=0.002$) on the same test used in our paper. We report the results in the following table.
>
> |Defense|Clean|Robust|
> |---|---|---|
> |None|100|3|
> |AudioPure|96|84|
> |[5] Baseline|98|13|
> |[5] Adv-Finetune-Joint-frozen|90|52|
> From the results, our method can still achieve the highest robust accuracy.

---

> ### Author Response · Authors · 2022-11-19
> **Response to Reviewer 6825 (Part 2)**
>
> >Q3. "Limitation" section is hastily written. It may be removed.
>
> A3.  Sorry for it. We will rewrite the "Limitation" section to improve the quality.
>
>
> >Q4. I could not understand the intuition behind the proposal. Why should diffusion models work for cleaning adversarial signals?
>
> A4. The intuition behind the proposed method is as follows: the role of the forward process of the diffusion model acts as to gradually remove the local structures of the data by adding noise. Given an adversarial example as the input, the perturbation will also be gradually smoothed via the forward process of the diffusion model so that the adversarial behavior has the potential to be removed. Then we apply the reverse process of the diffusion model to recover the clean input. In this way, the adversarial perturbation can be "washed out". Note that the time steps $n$ of the forward process of the diffusion model play an important role for the quality of "washed out (purification)". If $n$ is too small, the forward process of the diffusion model can not remove the adversarial behavior since it does not add enough noise to smooth the adversarial perturbation, resulting in low robust accuracy. While $n$ is too big, the forward process of the diffusion model will not only remove the adversarial behavior but smooth the original image semantic information so that the reversed examples will be another image different from the category of the original input. Thus, there exists a sweet point for the diffusion timestep $n$ for better robustness. Our Table 2 has shown the results.  We also provide a theoretical analysis in the Appendix D.
>
> Reference:
>
> [1] Naseer, Muzammal, et al. "A self-supervised approach for adversarial robustness." Proceedings of the IEEE/CVF Conference on Computer Vision and Pattern Recognition. 2020.
>
> [2] Samangouei, Pouya, Maya Kabkab, and Rama Chellappa. "Defense-GAN: Protecting Classifiers Against Adversarial Attacks Using Generative Models." International Conference on Learning Representations. 2018.
>
> [3] Donahue, Chris, Julian McAuley, and Miller Puckette. "Adversarial Audio Synthesis." International Conference on Learning Representations. 2018.
>
> [4] https://github.com/lukysummer/WaveGAN-Speech-Synthesis
>
> [5] Joshi, Sonal, et al. "Defense against Adversarial Attacks on Hybrid Speech Recognition using Joint Adversarial Fine-tuning with Denoiser." arXiv preprint arXiv:2204.03851 (2022).
>
> [6] Luo, Yi, and Nima Mesgarani. "Conv-tasnet: Surpassing ideal time-frequency magnitude masking for speech separation." IEEE/ACM transactions on audio, speech, and language processing 27.8 (2019): 1256-1266.
>
> [7] https://github.com/funcwj/conv-tasnet

---

> ### Author Response · Authors · 2022-11-28
> **Looking forward to further discussion**
>
> Dear Reviewer,
>
> We really appreciate your valuable comments. In our response, we have included additional experiments on DefenseGAN and Adversarial Fine-tuning. Besides, we have explained the intuition behind our method and improved our writing in the revised manuscript. We hope that your concerns have been addressed.
>
> As you might know that we are closer to the reviewer final recommendation deadline. Thus we would really appreciate it if you can provide us the further feedback on our detailed rebuttal. We have incorporated all the changes in our revised manuscript for your kind consideration.
>
> If you feel happy with our response, please consider updating your score. Feel free to reach us out in case you need any clarification.
>
> Paper4179 Authors

---

### Decision · Program_Chairs · 2023-01-20

**Decision:**

Accept: poster

**Justification For Why Not Higher Score:**

The work is a straightforward extension of DiffPure to the audio domain.

**Justification For Why Not Lower Score:**

The extension, while conceptually straightforward, requires nontrivial additional work and the authors do a comprehensive experimental evaluation of the approach.

**Metareview: Summary, Strengths And Weaknesses:**

The paper proposes using diffusion models for creating robust audio tasks. The key idea is to use diffusion models to "purify" a potentially adversarially corrupted audio signal and use the purified signal for downstream tasks. This is an extension of a similar approach applied to vision tasks, but is thoroughly validated in terms of both white/black box attacks and certified robustness achieved with randomized smoothing.

Strengths:
1. Comprehensive experimental evaluation across various datasets and types of attacks.
2. Extending use of diffusion models for adversarial robustness to audio domain.

Weaknesses:
1. Novelty: The work is an extension of existing proposed methods like DiffPure to the audio domain. While the extension is straightforward conceptually, the application to audio models is certainly a worthy contribution.

Hence, I recommend acceptance.

**Note From Pc:**

if the above contains the word "oral" or "spotlight" please see: "oral" presentation means -> notable-top-5% and "spotlight" means -> notable-top-25%. As stated in our emails, we are disassociating presentation type from AC recommendations

**Summary Of Ac-Reviewer Meeting:**

No meeting